# miR-515-5p controls cancer cell migration through MARK4 regulation

Olivier E Pardo[1],[†],[*], Leandro Castellano[1],[†], Catriona E Munro[1],[†], Yili Hu[2], Francesco Mauri[1], Jonathan Krell[1], Romain Lara[1], Filipa G Pinho[1], Thameenah Choudhury[1], Adam E Frampton[1], Loredana Pellegrino[1], Dmitry Pshezhetskiy[1], Yulan Wang[2], Jonathan Waxman[1], Michael J Seckl[1,3],[**] & Justin Stebbing[1,3],[***]

## Abstract

Here, we show that miR-515-5p inhibits cancer cell migration and metastasis. RNA-seq analyses of both oestrogen receptor receptor-positive and receptor-negative breast cancer cells overexpressing miR-515-5p reveal down-regulation of NRAS, FZD4, CDC42BPA, PIK3C2B and MARK4 mRNAs. We demonstrate that miR-515-5p inhibits MARK4 directly 3′ UTR interaction and that MARK4 knock-down mimics the effect of miR-515-5p on breast and lung cancer cell migration. MARK4 overexpression rescues the inhibitory effects of miR-515-5p, suggesting miR-515-5p mediates this process through MARK4 down-regulation. Furthermore, miR-515-5p expression is reduced in metastases compared to primary tumours derived from both *in vivo* xenografts and samples from patients with breast cancer. Conversely, miR-515-5p overexpression prevents tumour cell dissemination in a mouse metastatic model. Moreover, high miR-515-5p and low MARK4 expression correlate with increased breast and lung cancer patients' survival, respectively. Taken together, these data demonstrate the importance of miR-515-5p/MARK4 regulation in cell migration and metastasis across two common cancers.

**Keywords** breast cancer; lung cancer; microRNAs; microtubule affinity-regulating kinase 4; miR-515-5p

**Subject Categories** Cancer; Cell Adhesion, Polarity & Cytoskeleton; RNA Biology

## Introduction

Initially termed MARKL1 (microtubule affinity-regulating kinase-like 1), MARK4 was identified by a cDNA microarray approach to be down-regulated following a decrease in TCF/LEF1 (transcription complex, T-cell factor/lymphoid enhancer-binding factor) activity in hepatocellular carcinoma cell lines. To date, MARK4 expression has been found to be increased in hepatocellular carcinomas and gliomas, suggesting a role for MARK4 in cancer development [1,2]. Using a tandem affinity purification approach, MARK4 was found to interact with a number of proteins linked to the regulation of cell motility, namely 14-3-3 proteins, ARHGEF2 (GEF-H1)—a microtubule-associated exchange factor for Rho GTPases and phosphatase 2A (PP2A), and was previously shown to dephosphorylate TAU and other MAPs controlling their microtubule-binding affinity [3,4]. The identification of these binding partners and MARK4's interaction with three forms of tubulin (α-, β-, γ-tubulin), myosin and actin suggest a clear role for MARK4 in control of the cytoskeleton [5,6]. However, nothing is known about the control of MARK4 expression by, for example, microRNAs.

MicroRNAs (miRNAs) are small non-coding RNAs that post-transcriptionally regulate gene expression through binding to the 3′ UTR of their mRNA targets. In 2009, it was shown that more than 50% of known miRNA target genes are located in cancer-associated genomic regions (CAGRs), regions in the genome that are frequently altered in various cancers [7]. Therefore, unsurprisingly, the expression of miRNAs is often altered among different cancer sub-types and assessment of the expression of certain miRNAs has allowed for the accurate differentiation between benign and malignant tissues [8]. More recently, systemic injection of synthetic tumour suppressor miRNAs has been demonstrated to successfully prevent the growth of metastases in animal models with cancer metastasis [9]. This evidence suggests that miRNAs-based therapies could be used to treat metastatic disease or to prevent the formation of metastasis in early-stage disease.

miR-515-5p was initially described to be a placenta-specific miRNA involved in foetal growth [10]. However, recently, we identified its role as a tumour suppressor in breast cancer [11]. We

1 Department of Surgery & Cancer, Division of Cancer, Imperial Centre for Translational and Experimental Medicine (ICTEM), Imperial College, Hammersmith Hospital Campus, London, UK
2 Department of Oncology, Imperial College Healthcare NHS Trust, Charing Cross Hospital, London, UK
3 Key Laboratory of Magnetic Resonance in Biological Systems, Wuhan Center for Magnetic Resonance, State Key Laboratory of Magnetic Resonance and Atomic and Molecular Physics, Wuhan Institute of Physics and Mathematics, Chinese Academy of Sciences, Wuhan, China
*Corresponding author. Tel: +44 2075942814; E-mail: o.pardo@imperial.ac.uk
**Corresponding author. Tel: +44 2033111421; E-mail: m.seckl@imperial.ac.uk
***Corresponding author. Tel: +44 203 3118295; E-mail: j.stebbing@imperial.ac.uk
†These authors contributed equally to this work

demonstrated that miR-515-5p transcription was directly down-regulated by the oestrogen receptor and that miR-515-5p inhibited breast cell proliferation by inducing apoptosis [11]. During this work, a change in MCF-7 and MDA-MB-231 cell morphology was also observed, a finding that we investigate further here.

Indeed, we show that miR-515-5p overexpression increases microtubule area not only in breast cancer but also in non-small cell lung cancer (NSCLC) cells and that this correlates with decreased cell migration. We performed an RNA-seq following the overexpression of miR-515-5p in MCF7 and MDA-MB-231 cells, which revealed the down-regulation of five transcripts linked to cell migration: NRAS, FZD4, CDC42BPA, PIK3C2B and MARK4 [6,12–15]. These four transcripts were also down-regulated in A549 and H1299 NSCLC cells in response to miR-515-5p overexpression and luciferase reporter constructs, indicating that miR-515-5p directly regulated NRAS, PIK3C2B and MARK4 expression in multiple cancer cells. As the strongest down-regulation of expression by miR-515-5p was seen with MARK4 in both MCF7 and MDA-MB-231 and that this gene had previously been reported to modulate NSCLC cell motility [16], we decided to focus our study on investigating the direct regulation of MARK4 expression by miR-515-5p. We showed that MARK4 silencing reproduces the morphological and cell migration changes observed following miR-515-5p overexpression in both breast and NSCLC cells, demonstrating this to be a consistent phenomenon across two separate tumour types. Conversely, transfection of exogenous MARK4 into miR-515-5p overexpressing cells rescued the observed decrease in cell migration, suggesting that MARK4 down-regulation is a crucial mechanism through which miR-515-5p reduces cell motility. By analysing miR-515-5p expression in tumour samples derived from breast cancer patients, we observed its expression to be inversely correlated with metastasis, while expression of miR-515-5p mimics in A549 NSCLC cells prevented *in vivo* tumour cell dissemination in a metastatic mouse model. In agreement with metastasis being a poorer prognosis factor for cancer patients, decreased miR-515-5p or increased MARK4 expression was indicative of poorer survival in both metastatic breast and lung cancer. In short, our data demonstrate that miR-515-5p dramatically inhibits cell migration by directly down-regulating MARK4 expression in two different cancer types and suggests a role for miR-515-5p and MARK4 as potential biomarkers in metastatic disease and as possible therapeutic targets.

# Results

## miR-515-5p changes cell morphology and inhibits cell migration in breast cancer

We initially overexpressed miR-515-5p in breast (MCF7 and MDA-MD-231) and lung (A549 and H1299) cell lines (Appendix Fig S1A) and stained for tubulin to assess cell morphology. We observed an increase in cell area and loss of cell polarity (Fig 1A). The loss of cell polarity was confirmed quantitatively by comparing the cell circularity factor of individual cells in 10 fields of view between control and miR-515-5p overexpressing conditions for the four cell lines (Appendix Fig S2). Overexpression of miR-515-5p significantly increased the circularity factor towards a value of 1, indicating loss of cell elongation.

As cell cytoskeleton dynamics and cell polarity are crucial for cell motility, we wished to investigate whether this change in morphology affected cancer cell migration. We analysed the effect of miR-515-5p overexpression in MDA-MB-231, A549 and H1299 on random and directional migration by conducting random and Boyden chamber directional cell migration assays. We observed a sharp decrease in random and directional cell migration in miR-515-5p-transfected cells as compared to their miRVANA non-targeting controls (NC) (Fig 1B and C). This however was not the result of overexpression of this miRNA inducing apoptotic cell death as shown in Appendix Fig S1B for A549 and H2199 cells and previously published by us for MDA-MB-231 [11]. Hence, miR-515-5p overexpression is able to directly suppress cell migration in these cell lines possibly as a consequence of the observed changes to the cell cytoskeleton.

## miR-515-5p directly regulates NRAS, MARK4 and PIK3C2B expression

To identify which miR-515-5p targets were responsible for its effect on cancer cell migration, we performed an RNA-seq analysis of MCF7 and MDA-MB-231 cells overexpressing miR-515-5p. Interestingly, we found 5 down-regulated transcripts which were predicted to interact with miR-515-5p and have also been implicated in cell migration: NRAS, FZD4, CDC42BPA, PIK3C2B and MARK4 (Fig 2A, Dataset EV1) [6,12–15]. To validate the RNA-seq results, we performed validatory qPCRs for the levels of these five transcripts in MCF7 and MDA-MB-231 cells upon the overexpression of miR-515-5p (Fig 2B and C) and miR-515-5p sponge vectors (Appendix Fig S3A and B), which reduce the levels of miR-515-5p by directly interacting with its mature form. miR-515-5p sponge vectors induced an increase in the levels of the 5 transcripts in MCF7 but not within MDA-MB-231 cells (Appendix Fig S3A and B). This is likely to be because miR-515-5p expression is much lower in MDA-MB-231 than in MCF7 (Appendix Fig S4). Conversely, we observed a significant down-regulation of the expression of all five transcripts in the miR-515-5p-transfected MCF7 cells but only a decrease in N-RAS, PI3KC2B and MARK4 mRNA levels in miR-515-5p-transfected MDA-MB-231 (Fig 2B and C). Interestingly, miR-515-5p dramatically decreased MARK4 mRNA expression (> 95%) in both MCF7 and MDA-MB-231 cells ($P < 0.001$; Fig 2B and C). Levels of these five transcripts were also decreased in NSCLC cells (A549 and H1299) upon overexpression of miR-515-5p (although not significantly for CDC42BPA in A549 cells), underlying a common molecular function for this miRNA across tumour types (Fig 2D).

We then wished to assess whether the transcripts for N-RAS, MARK4 and PI3KC2B could be directly regulated by the binding of miR-515-5p to their 3′ UTRs. We used a reporter system where luciferase expression was under the control of the 3′ UTRs of our proposed targets. We showed that miR-515-5p negatively regulated the levels of luciferase reporter expression in both MCF7 and MDA231 cells by directly interacting with the 3′ UTR regions of N-RAS, MARK4 and PI3KC2B ($P < 0.001$; Fig 2D and E). This demonstrated that the mRNAs for these three proteins were true direct targets of miR-515-5p. The specificity of the binding of miR-515-5p to the 3′ UTR regions of MARK4 was demonstrated through mutating both the sites predicted by TargetScan in the seed sequence. Indeed, mutation of both sites in the 3′ UTR prevented miR-515-5p from inhibiting luciferase expression from our reporter vector (Appendix Fig S5).

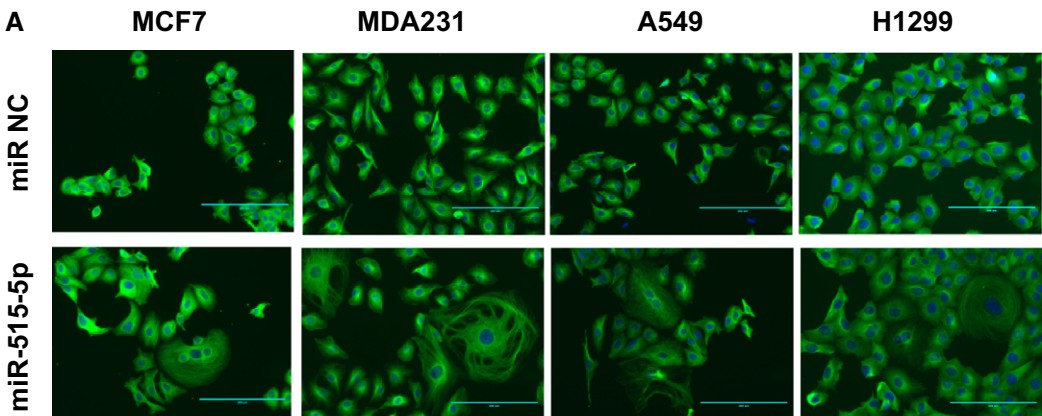

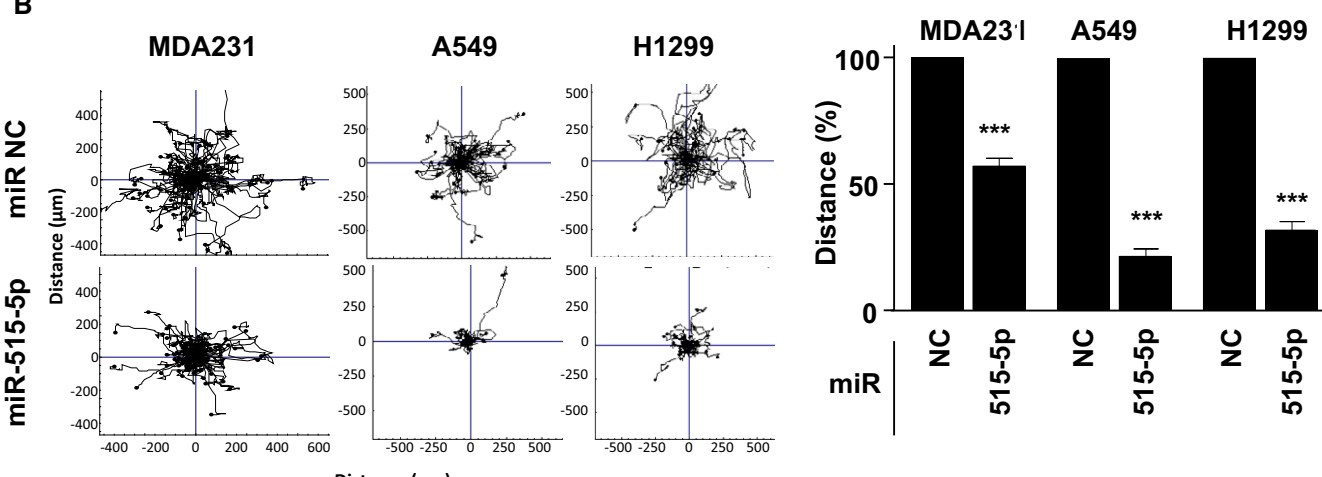

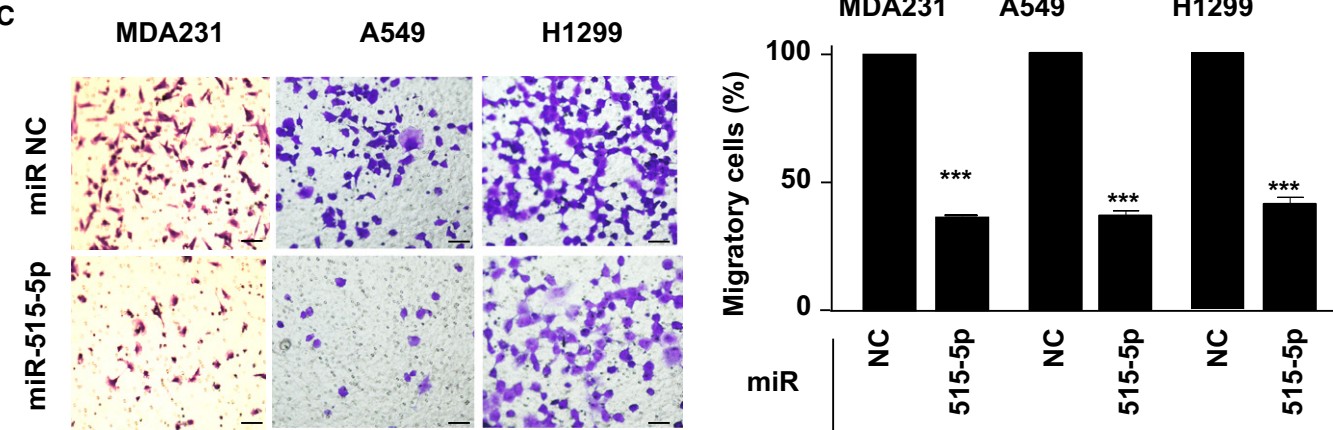

**Figure 1.  miR-515-5p expression changes the tubulin cytoskeleton and inhibits migration of breast and lung cancer cells.**

A   miR-515-5p expression increases the area of the tubulin cytoskeleton. Tubulin (green) and cell nucleus (blue). Objective x20. Scale bar: 200 μm.

B, C   miR-515-5p inhibits random (B) and directed (C) cell migration. The indicated cell lines were transfected with miR-515-5p for 48 h before time-lapse imaging was performed for 18 h (A), or transwell migration assays were performed for 9 h (C). (B, left panel) Plots show overlays of representative trajectories travelled. (B, right panel) The distance of migration was quantified and represented as the mean ± SEM of values normalised to the respective control condition. $n \geq 30$ cells tracked per condition. (C, left panel) Representative field of view from the bottom of the transwell chambers. Scale bar is 20 μm. (C, right panel) Data represent percentage of the average number migratory cells per field ($n = 5$ fields per condition). Data are mean of three experiments ± SEM. *P*-values were calculated by *t*-test between miRNA (515-5p) values and their respective non-targeting control miR (NC) values (***$P < 0.001$). All data shown are representative of experiments performed at least in triplicate.

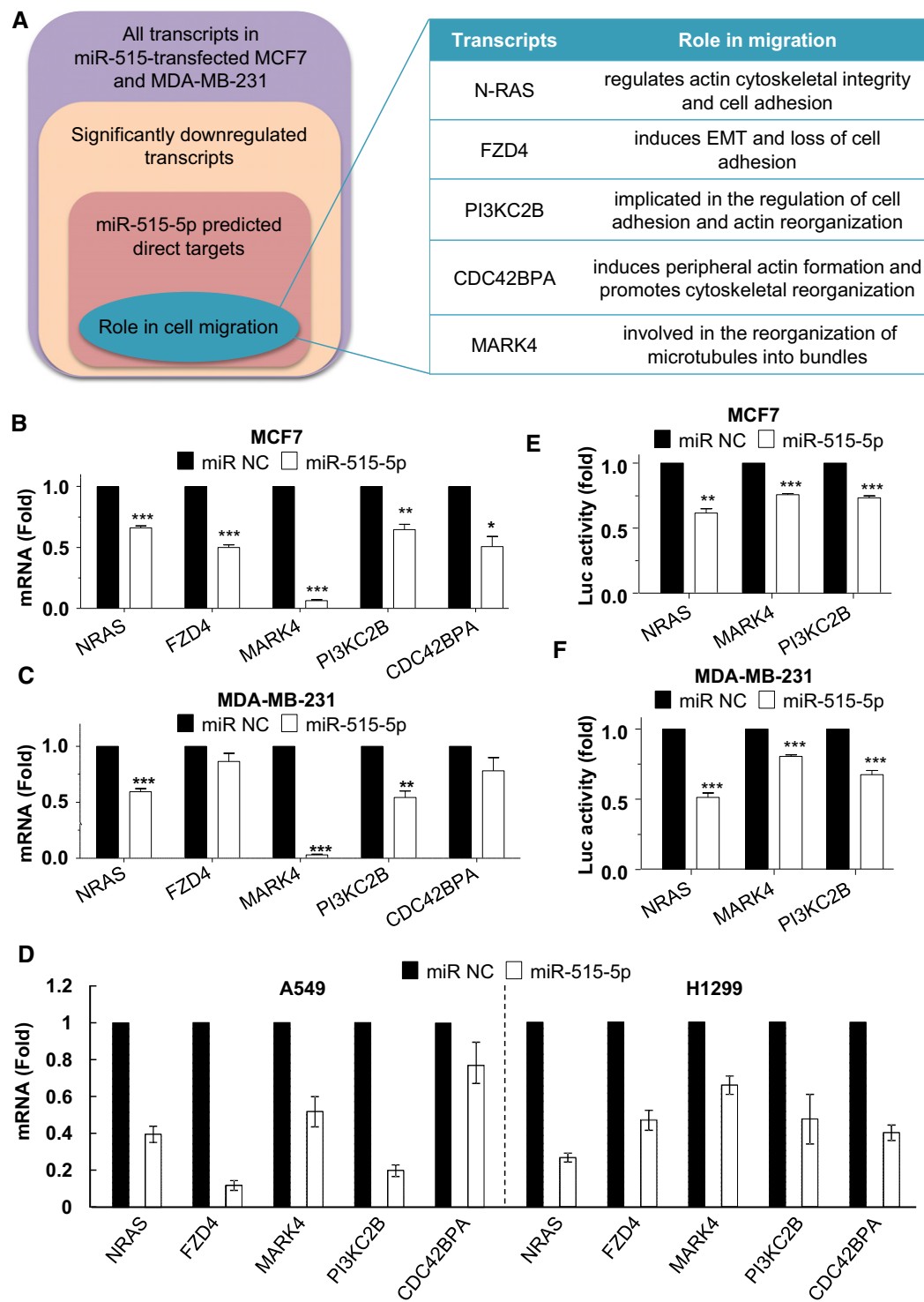

**Figure 2.  miR-515-5p regulates mRNAs involved in cell migration.**

A      RNA-seq of MCF7 and MDA-MB-231 transfected with miR-515-5p revealed the down-regulation of five transcripts, NRAS, FZD4, CDC42BPA, PIK3C2B and MARK4.

B–D   The effect of miR-515-5p overexpression on NRAS, FZD4, CDC42BPA, PIK3C2B and MARK4 mRNA levels in MCF7 (B), MDA-MB-231(C), A549 and H1299 (D) was determined by qPCR 48 h following transfection of the indicated cell line with non-targeting control miR (NC) or miR-515-5p. Data are displayed as the normalised mean ± SEM of $n = 4$. $P$-values were calculated by $t$-test between miRNA conditions and their respective NC conditions (*$P < 0.05$; **$P < 0.01$; ***$P < 0.001$).

E, F    miR-515-5p directly interacts with NRAS, MARK4 and PI3KC2B's 3′ UTR. Relative luciferase activity levels were measured 24 h after co-transfection of MCF-7 (E) and MDA-MB-231 (F) with 3′ UTR-luciferase reporter constructs and either with miR-515-5p or miR-NC. Data shown are normalised mean of three independent experiments ± SEM.

Data information: (B–F) $P$-values were calculated by $t$-test between miRNA conditions and their respective NC conditions (*$P < 0.05$; **$P < 0.01$; ***$P < 0.001$).

## Silencing of MARK4 mimics the effect of miR-515-5p on both random and directional migration

As MARK4 was so predominantly down-regulated by miR-515-5p in breast cancer cells (Fig 2B and C), we questioned whether silencing MARK4 would mimic the effect of miR-515-5p on cell morphology and in cell migration assays. As MCF7 cells are known to display poor motility, we did not analyse the effect of either miR-515-5p or MARK4 silencing on MCF7 cell migration [17]. Hence, we limited our primary analysis to MDA-MB-231, A549 and H1299 cell lines. Transfection of these cell lines with MARK4-targeting siRNAs led to an increase in the cell area and clustering as can be observed following tubulin staining (Fig 3A). The change in cell area (quantified in Fig 3A, right panel) was accompanied by a loss of cell polarity reminiscent of that observed for miR-515-5p transfection (Fig 1A). We therefore proceeded to study possible associated changes in cell motility. Our previously published cell migration screen had revealed that silencing of MARK4 reduced migration in A549 cells [16], so we used this cell line as an internal control in these assays. Random migration (RM) and Boyden chamber directional migration (DM) assays showed that MARK4 knock-down inhibited cell migration by 48–61%, 55–58% and 54–56% (RM-DM) in MDA-MB-231, A549 and H1299 cell migration, respectively (Fig 3B and C). To further validate the role of MARK4 in the migration of breast cancer cells, we performed random migration experiments in migratory MDA-157 breast cancer cells and found that MARK4 silencing significantly reduced their motility (Appendix Fig S6B). A similar effect was obtained in these cells upon overexpression of miR-515-5p (Appendix Fig S6B) and was accompanied by changes in cell morphology comparable to that observed in MDA-MB-231, A549 and H1299 cells (Appendix Fig S6C). Taken together, these data suggest that MARK4 down-regulation might be the major cause for the decrease in cell migration observed upon miR-515-5p overexpression.

## miR-515-5p inhibits cell migration through MARK4 down-regulation

To clarify the importance of MARK4 in the cell migration inhibition by miR-515-5p, we transfected MDA-MB-231, A549 and H1299 cells with miR-515-5p, and after 24 h, we overexpressed MARK4 in the transfected cell lines (Appendix Fig S7A). We then analysed whether MARK4 overexpression rescued the effect of miR-515-5p on the random and directional migration of these cell lines (Fig 4A and B). There was no significant difference between the distance travelled by MDA-MB-231, A549 and H1299 cells co-overexpressing miR-515-5p and MARK4 and that of cell lines co-transfected with miR-NC and empty vector control plasmid (EV) in the random migration assays (Fig 4A and Appendix Fig S7B). Similarly, in the directional assay, the expression of MARK4 rescued the decrease in MDA-MB-231, A549 and H1299 cell migration, although this was more prominent for the lung cancer cell lines (Fig 4B and Appendix Fig S7C).

## MARK4 silencing triggers cell cycle arrest

In addition to the changes in cell motility, regulation of microtubule dynamics would also be expected to have significant effects on cell division. We therefore compared the cell cycle distribution between control-transfected A549 cells and cells silenced for MARK4. This

analysis revealed that MARK4-down-regulated cells accumulated in the G1 phase of the cell cycle while showing a decrease in DNA synthesis (Appendix Fig S8A). This was accompanied by an increase in the levels of p53 as well as its transcriptional target, p21, in MARK4-silenced cells, suggesting that down-regulating this kinase induces cell cycle arrest (Appendix Fig S8B). To analyse the speed of microtubule growth, a plasmid encoding GFP tagged end-binding protein 3 (EB3) was transfected into A549 cells. This protein binds to the growing end of the microtubules (MT), and by imaging the movement of the GFP comets produced as the microtubule grows, we were able to quantify the speed of growth and compare this growth rate between cells silenced for MARK4 or transfected with a non-targeting control siRNA. Representative images of the comets for the two conditions can be seen in Appendix Fig S9A. When quantified using ImageJ and the wrMTrck plug-in, the comet speed was found to be significantly higher in cells silenced for MARK4 suggesting an increase in the rate of tubulin polymerisation (Appendix Fig S9B).

## MARK4 expression levels increase during tumorigenesis and are inversely correlated to patient survival

To assess the clinical relevance of our findings, we first evaluated the potential of MARK4 as prognostic biomarkers in metastatic breast and lung cancer patients (Fig 5).

We analysed the overall survival (months) of a cohort of patients with metastatic lung or breast cancer and correlated this with the levels of MARK4 mRNA present in their tumours. High levels of MARK4 mRNA correlated with a lower overall survival in ER-negative metastatic breast and lung cancer patients, although this only reached statistical significance in the latter (Fig 5A and B). Quantitative PCR analysis of a lung cancer cDNA array revealed that MARK4 mRNA levels are increased in > 50% of tumours as compared to normal lung tissues (Fig 5C). In agreement with this, MARK4 protein was detectable in 37% of lung cancer samples, while it remained undetected in normal lung tissue (Fig 5D). When subtypes of lung cancer were separated, MARK4 expression was especially prevalent in adenocarcinoma samples with 53% of cases for this tumour type showing positive MARK4 staining (Fig 5E). In addition, comparison of metastatic and primary lesions form the same patients revealed that MARK4 expression was increased in secondary as compared to primary tumours in 41% of cases (Fig 5F). Hence, these data suggest that MARK4 expression increases during tumorigenesis, impacting negatively on patients' survival.

Taken together, MARK4 and/or miR-515-5p expression levels may be relevant prognosis markers in both breast and lung cancer.

## miR-515-5p expression is inversely correlated with metastasis in breast and lung cancer mouse models

In contrast to MARK4, miR-515-5p expression was positively correlated with overall survival in patients with metastatic breast or lung cancer (Fig 6A–D and Appendix Fig S10A and B). Moreover, in luminal B breast cancer and in normal breast epithelial cells, miR-515-5p levels were significantly inversely correlated with MARK4 expression and a similar trend existed in the case of luminal A and basal breast cancer (Appendix Fig S11). Because the advent of metastatic disease is one of the main factors that negatively impact on patients' survival, we proceeded to measure miR-515-5p levels in

    

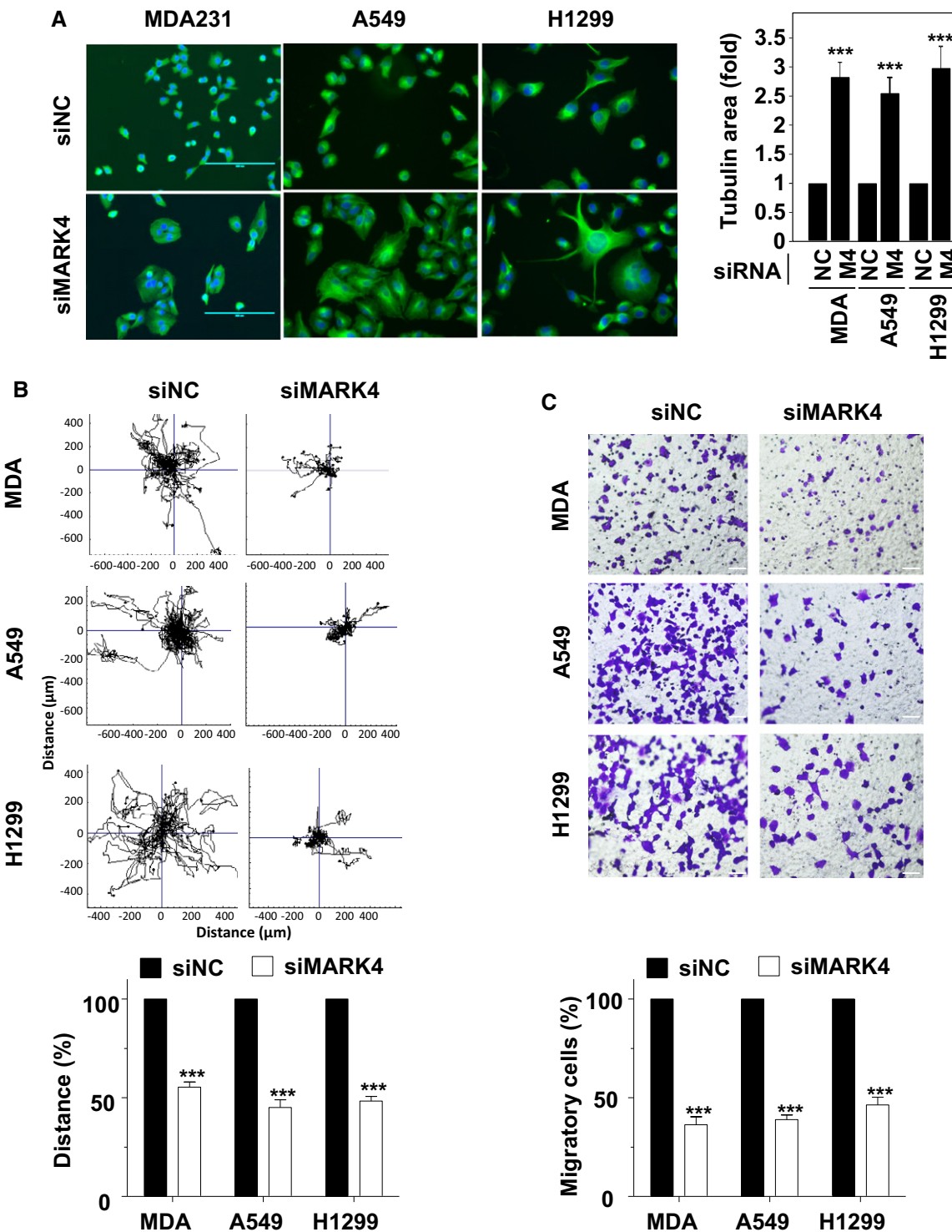

**Figure 3.  MARK4 knock-down mimics the effect of miR-515-5p overexpression.**

A–C  MARK4 siRNA-mediated silencing increases the tubulin cytoskeletal area (A) and reduces random (B) and directed (C) migration in MDA-MB-231 (MDA), A549 and
H1299 cells. (A) Data shown are representative of experiments performed in triplicate. Bar graph represents the average ± SEM of $n \geq 50$ per technical replicate
with three technical replicates per conditions. Cells were transfected either with siRNA targeting MARK4 (M4) or a non-targeting control (NC) for 48 h before time-
lapse imaging was performed for 18 h (B) or transwell assay for 9 h (C). (B, top panel) Plots show overlays of representative trajectories described. (B, bottom panel)
The migration distance was quantified and represented as the normalised mean ± SEM. (C, top panel) Representative field of view from the bottom side of the
transwell membrane. (C, bottom panel) Graphs indicate cell migration expressed as a percentage of the average of migratory cells per field ($n$ = 5 fields per
condition). (B–C) Data are mean of three experiments ± SEM. (A–C) $P$-values were calculated by $t$-test between the siRNA-transfected and the respective control
condition (***$P$ < 0.001). Scale bar: 200 μm.

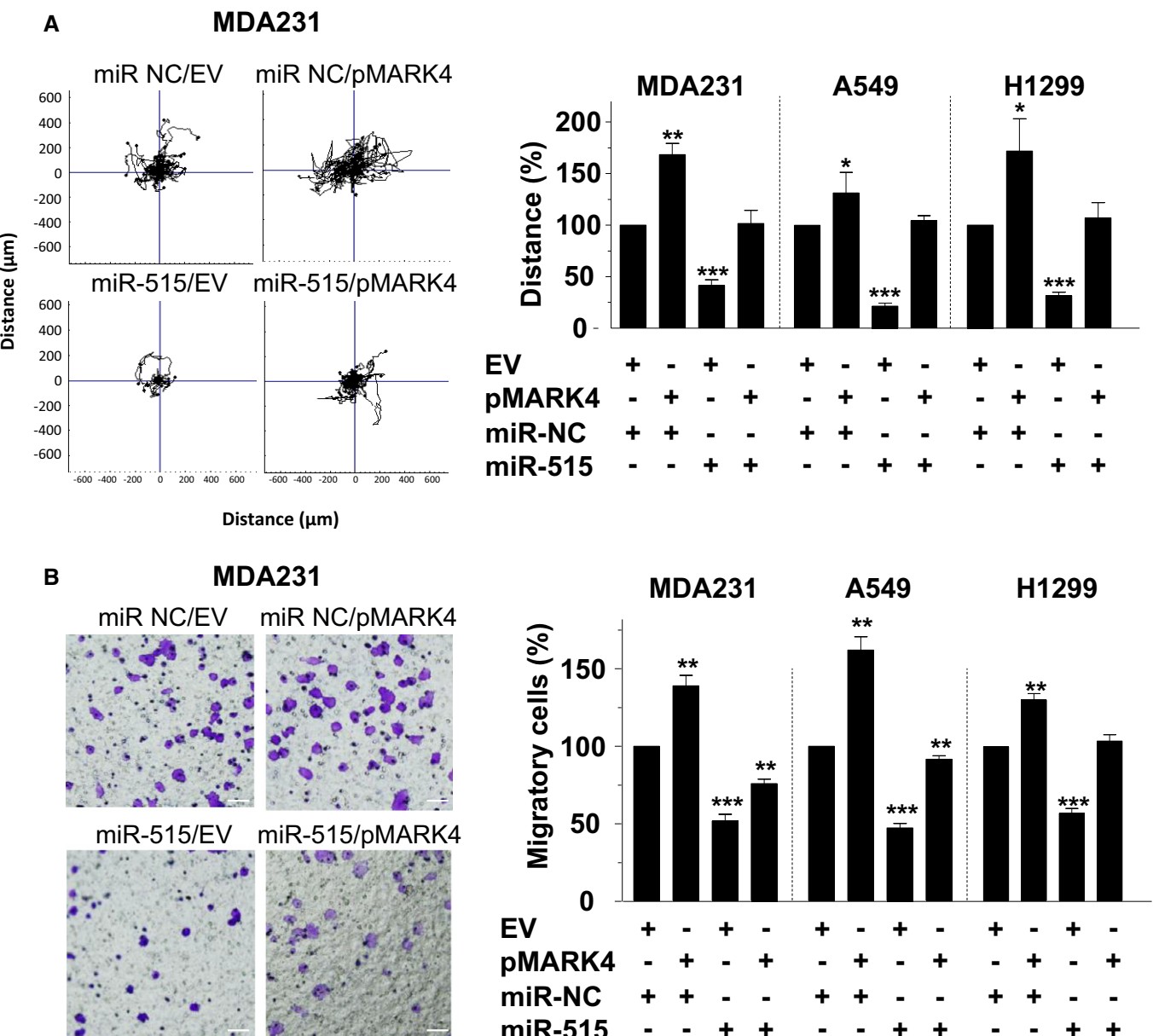

**Figure 4. MARK4 overexpression rescues the inhibitory effects of miR-515-5p on cell migration.**

A, B    The indicated cell lines were transfected with either miR515-5p or a non-targeting miR (miR-NC) together with a MARK4 expressing or an empty vector (EV) plasmid DNA. MARK4 overexpression rescues the inhibition by miR-515-5p of random (A) and directed (B) migration of the indicated cell lines. (A, left panel) Plots show overlays of representative trajectories described for MDA-MB-231 cells (*n* = 30). (A, right panel) The migration distance was quantified and represented as the normalised mean ± SEM. (B, left panel) Representative field of view from the bottom side of the transwell membrane for MDA-MB-231. (B, right panel) Graphs indicate cell migration expressed as a percentage of the average of migratory cells per field (*n* = 5 fields per condition). Data are mean of three experiments ± SEM. *P*-values were calculated by *t*-test between the transfected and the respective control condition (EV+miR-NC) (*$P$ < 0.05; **$P$ < 0.01; ***$P$ < 0.001). Scale bar: 200 μm.

paired primary and metastatic breast cancer patient samples. This demonstrated a lower mean expression of miR-515-5p in lymph node metastases compared to their corresponding primary tumours (Fig 6E), either by comparing the absolute values (Fig 6E) or the paired values for each patient (Appendix Fig S12).

To evaluate whether changes to the expression of miR-515-5p directly modulated the metastatic potential of breast and lung cancer cells, we performed two animal experiments. MDA-MB-231 cells

have previously been used to induce metastatic lesions in nude mice following inoculation into their mammary fat pads [19]. Using samples from both primary and metastatic lesions obtained in these animals, we showed that miR-515-5p levels were down-regulated in all metastatic lesions regardless of site (lung, bone, brain or adrenal glands) as compared to the corresponding primary tumour (Fig 6F, left panel). This was accompanied by a corresponding increase in MARK4 mRNA levels at the metastatic sites (Fig 6F, right panel).

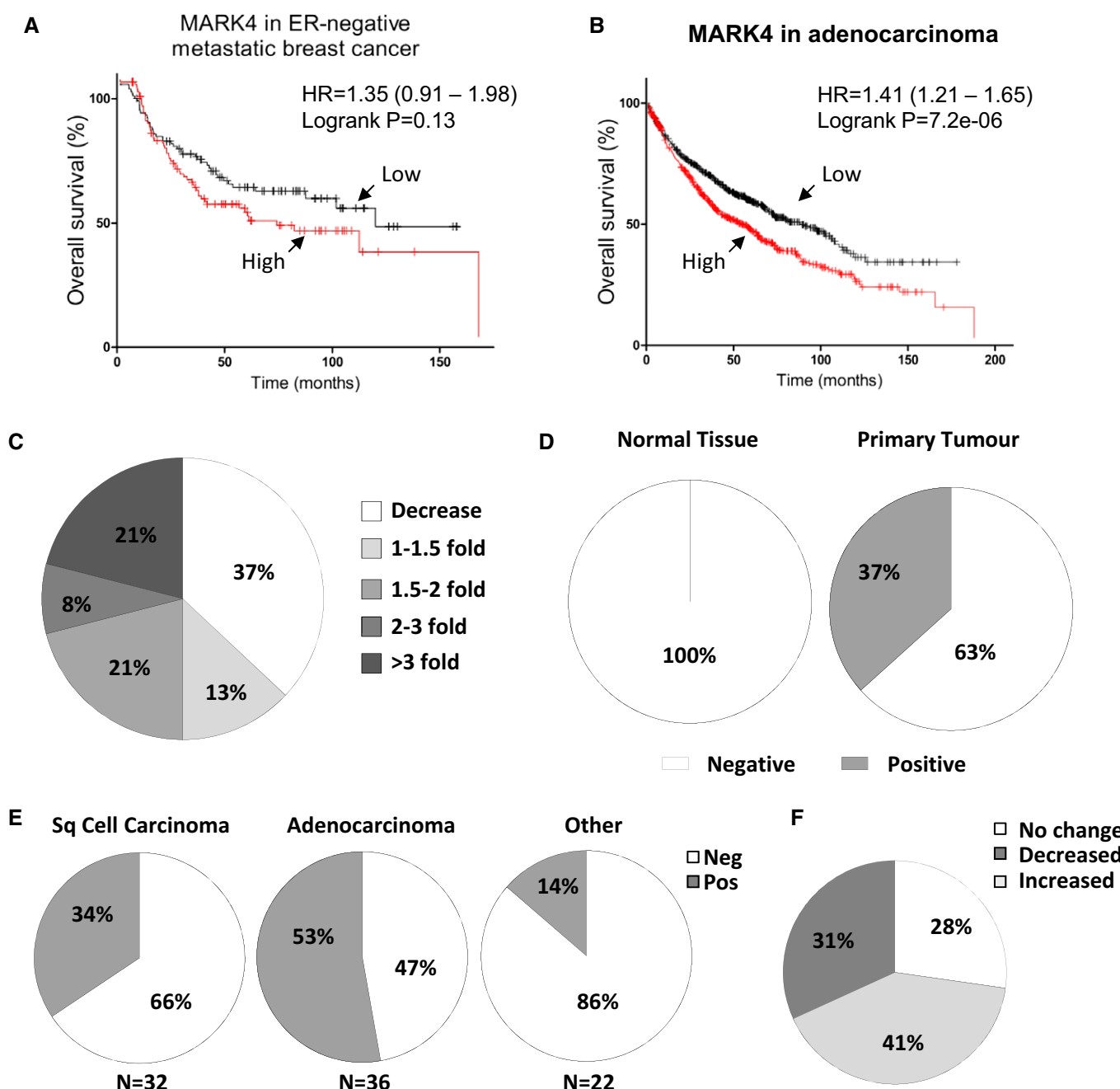

**Figure 5. MARK4 expression levels increase during tumorigenesis and correlate poor patient prognosis.**

A, B   Overall survival of ER-negative metastatic breast cancer (*n* = 226) (A) and lung adenocarcinoma (*n* = 719) (B) patients according to MARK4 mRNA levels found in their tumours was represented using the Kaplan–Meier plotter [25].

C   A cDNA array for 24 matched lung cancers and surrounding normal tissue was analysed for MARK4 expression. Expression levels from tumours were normalised to their respective normal control and the percentage of cases for the indicated fold expression in the tumour represented as a pie chart.

D–F   MARK4 protein expression was determined by histoimmunochemistry in (D–E) a tissue microarray comprising 90 lung tumour and 10 normal lung samples or (F) a home-made tissue microarray comprising 100 matched cases of primary and metastatic tumours. (D–E) The percentage of positivity for MARK4 staining in lung tumours versus normal lung samples (D) or in lung tumours according to their histological subtypes (E) is represented. (F) The percentage of cases with increased, decreased or unchanged levels of MARK4 in metastatic lesions as compared to the corresponding primary tumours is represented.

Interestingly, these changes did not necessarily extend to other miR-515-5p targets, as NRAS expression levels did not vary significantly between primary and metastatic lesions (Appendix Fig S13A)

and PI3KC2B expression was only significantly up-regulated in the metastatic cells isolated from brain and adrenal glands (Appendix Fig S13B). Hence, confirming the results of the rescue

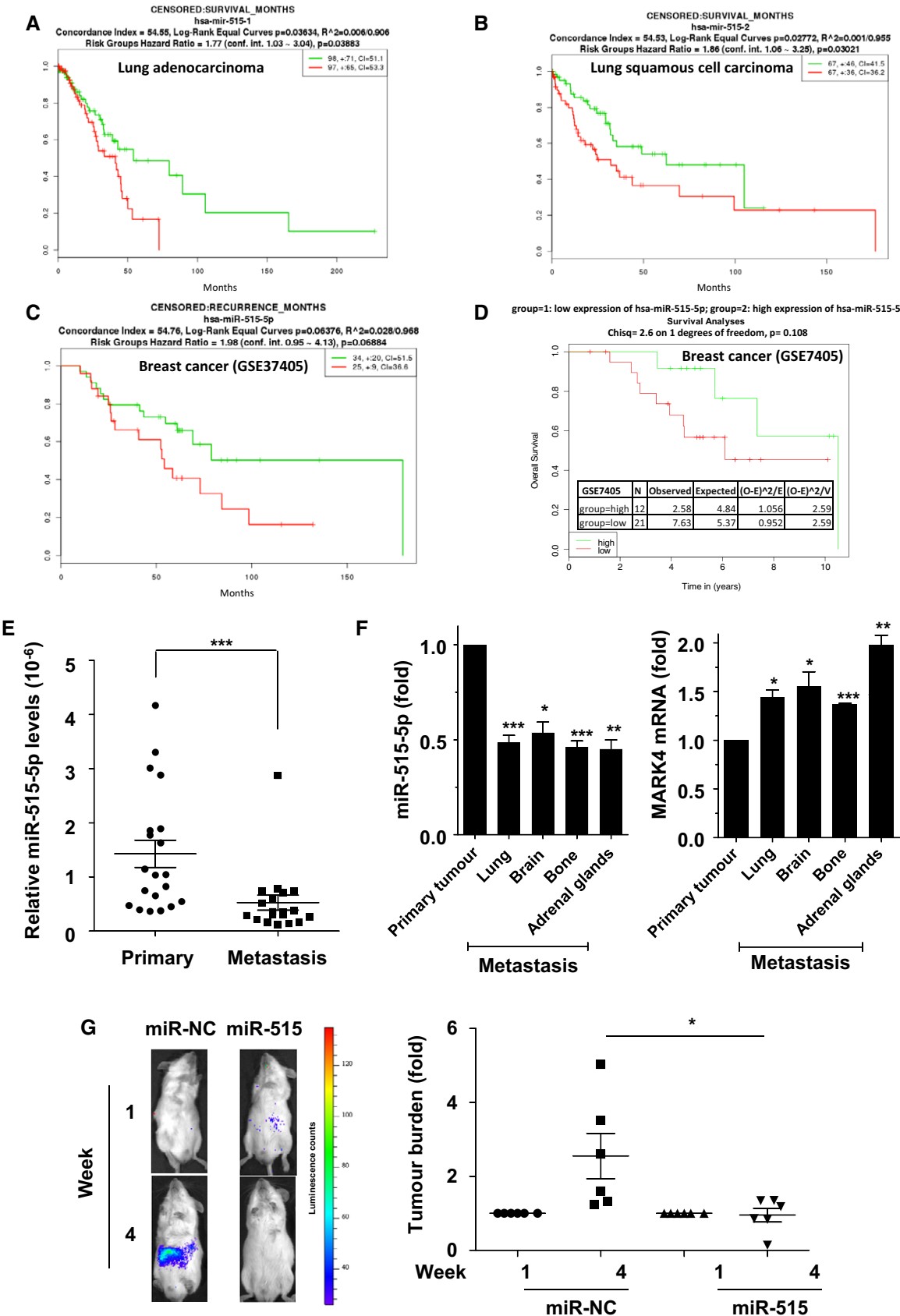

**Figure 6.**

**Figure 6.  miR-515-5p expression is inversely correlated with breast cancer metastasis and correlates with good patient prognosis.**

A–D    Low expression of miR-515-p is linked to survival in both lung and breast cancer. (A) Survival analysis of miR-515-5p from 195 lung adenocarcinoma samples from the TCGA. (B) Survival analysis of miR-515-p from 34 lung squamous cell carcinoma from the TCGA [18]. (C) Survival analysis of miR-515-5p from 60 breast cancer samples from the Lyng Ditzel Breast Study (GSE37405). (D) Survival analysis of miR-515-5p from 33 breast cancer from the GSE7405 Breast study. Analysis and plot were obtained with SurvMicro http://bioinformatica.mty.itesm.mx:8080/Biomatec/Survmicro.jsp for (A–C) and with MIRUMIR for (D).

E    Comparison of miR-515-5p expression in 18 primary and matched metastatic lesions from breast cancer patients. miR-515-5p levels were quantified by TaqMan qRT–PCR. Absolute values were normalised to U6 snRNA values and displayed as a dot-plot. Bar: mean; whiskers, SEM.

F    MDA-MB-231 cells were inoculated into the mammary fat pads of nude mice and allowed to form metastatic deposits [19]. miR-515-5p and MARK4 levels were then determined in the primary and indicated secondary lesions. Data are average $\pm$ SEM of technical triplicates.

G    Luciferase-expressing A549 cells overexpressing miR-515-5p or a non-targeting control (miR-NC) were injected in the tail vein of SCID mice and the development of tumour burden monitored by whole-animal imaging in response to luciferin administration at week 1 and week 4 following injection. (G, left panel) Representative imaging of the same animals for the indicated conditions at week 1 and week 4. Dots colors correspond to luminescence counts as per the scale on the right. (G, right panel) The emitted photons were quantified and the signals normalised to the corresponding readings at week 1. Data represented are representative of three independent experiments.

Data information: (E–G) *P*-values were calculated by *t*-test (\**P* < 0.05; \*\**P* < 0.01; \*\*\**P* < 0.001).

## Discussion

experiments, these data suggest that MARK4 may be the essential target regulating metastasis downstream of miR-515-5p (Fig 4A and B). While the preceding results suggested that decreased miR-515-5p expression was associated with the onset of metastasis, they did not demonstrate this change to be a causal phenomenon. Therefore, we wished to examine whether directly changing levels of this miRNA could modulate tumour cells colonisation *in vivo*. A549 cells stably expressing a luciferase reporter gene were transfected with or without miR-515-5p and the cells injected 72 h later into SCID mice through the tail vein. The animals were then monitored by whole-animal imaging following the administration of luciferin at 4 days and 4 weeks following the injection (Fig 6G, left panel). The bioluminescent signal was then quantified and plotted to show the extent of the metastatic burden (Fig 6G, right panel). This analysis revealed that while cells transfected with a non-targeting miRNA mimetic (miR-NC) efficiently colonised the mice, miR-515-5p-transfected cells failed to establish tumours in the injected animals (Fig 6G). Hence, changes in miR-515-5p levels alone are sufficient to modulate the metastatic potential of cancer cells.

### MARK4 silencing prevents metastatic lung colonisation of A549 cells

We next tested whether MARK4 silencing alone could reproduce the effects of miR-515-5p expression on the metastatic behaviour of A549 cells. A549-Luc cells were transfected with MARK4 siRNAs or a non-targeting sequence and injected 48 h later into SCID mice through the tail vein. Four weeks later, the mice were sacrificed and their lungs extracted and fixed prior to immunohistochemical staining for luciferase expression. As an internal positive control, we performed the same experiments using cells overexpressing miR-515-5p or a corresponding non-targeting sequence. Figure 7A shows that, 4 weeks following tail vein injection, cells transfected with non-targeting siRNA had colonised the lungs of the animals as revealed by the presence of luciferase-positive cells either as single cells (Fig 7A, upper panels) or as tight tumours (Fig 7A, lower panels). Quantification of the percentage of area of the fields of view covered by luciferase-positive cells revealed that MARK4 silencing, just as miR-515-5p overexpression, impaired lung colonisation by A549 cells (Fig 7B). These data further confirm that MARK4, as a downstream target of miR-515-5p, controls the invasive potential of cancer cells.

Having previously demonstrated the role of miR-515-5p as an ER-positive breast cancer tumour suppressor [11], here we show that miR-515-5p inhibits cell migration via its targeting of MARK4. Initially, we investigated the role of miR-515-5p in cell migration because we observed rearrangement of the tubulin cytoskeleton in MCF7 and MDA-MB-231 cells that overexpressed miR-515-5p (Fig 1A). We previously demonstrated that ERα suppresses miR-515-5p expression by direct interaction with its promoter, indicating a link between this miRNA and oestrogen signalling [11]. Interestingly, the expression of miR-515-5p is lower in MCF7 epithelial-like cells than in MDA-MB-231 which are more mesenchymal-like. This suggests that ERα is not the only transcription factor regulating miR-515-5p expression, but additional regulators modulate its levels in association with the mesenchymal phenotype. Both MCF7 and MDA-MB-231 cells appeared enlarged with a loss of polarity that is often associated with non-migratory cells. This change did not correlate with the onset of cell death downstream of this miRNA [11] indicating that the effects of miR-515-5p on cell morphology may be caused by distinct targets and, consequently, through different signalling pathways. Additionally, because MDA-MB-231 is an ER-negative breast cancer cell line, we propose that the oestrogen receptor-α is not involved in these effects. Moreover, miR-515-5p caused similar cytoskeletal changes in non-small cell lung cancer (NSCLC) cells, highlighting a common function of this miRNA across tumour types.

As altered cytoskeletal dynamics and cell polarity are known to influence cell motility, we investigated the effect of miR-515-5p on cell migration. Consistent with the morphological changes (Fig 1A and Appendix Figs S2 and S6C), miR-515-5p inhibited both random and directional cell migration in both breast and lung cancer cell lines (Fig 1B and C and Appendix Fig S6B). In order to ascertain the mechanism through which miR-515-5p altered cell morphology and migration and identify its gene targets, we performed an RNA-seq analysis using RNA obtained from miR-515-5p-transfected MCF7 and MDA-MB-231 cells. We identified 5 transcripts down-regulated upon miR-515-5p overexpression: NRAS, FZD4, CDC42BPA, PIK3C2B and MARK4 (Fig 2A), and these were validated by qPCR (Fig 2B and C). These transcripts have been described as being involved in cell migration and are predicted by TargetScan to be directly targeted by miR-515-5p [6,12–15]. Furthermore, we

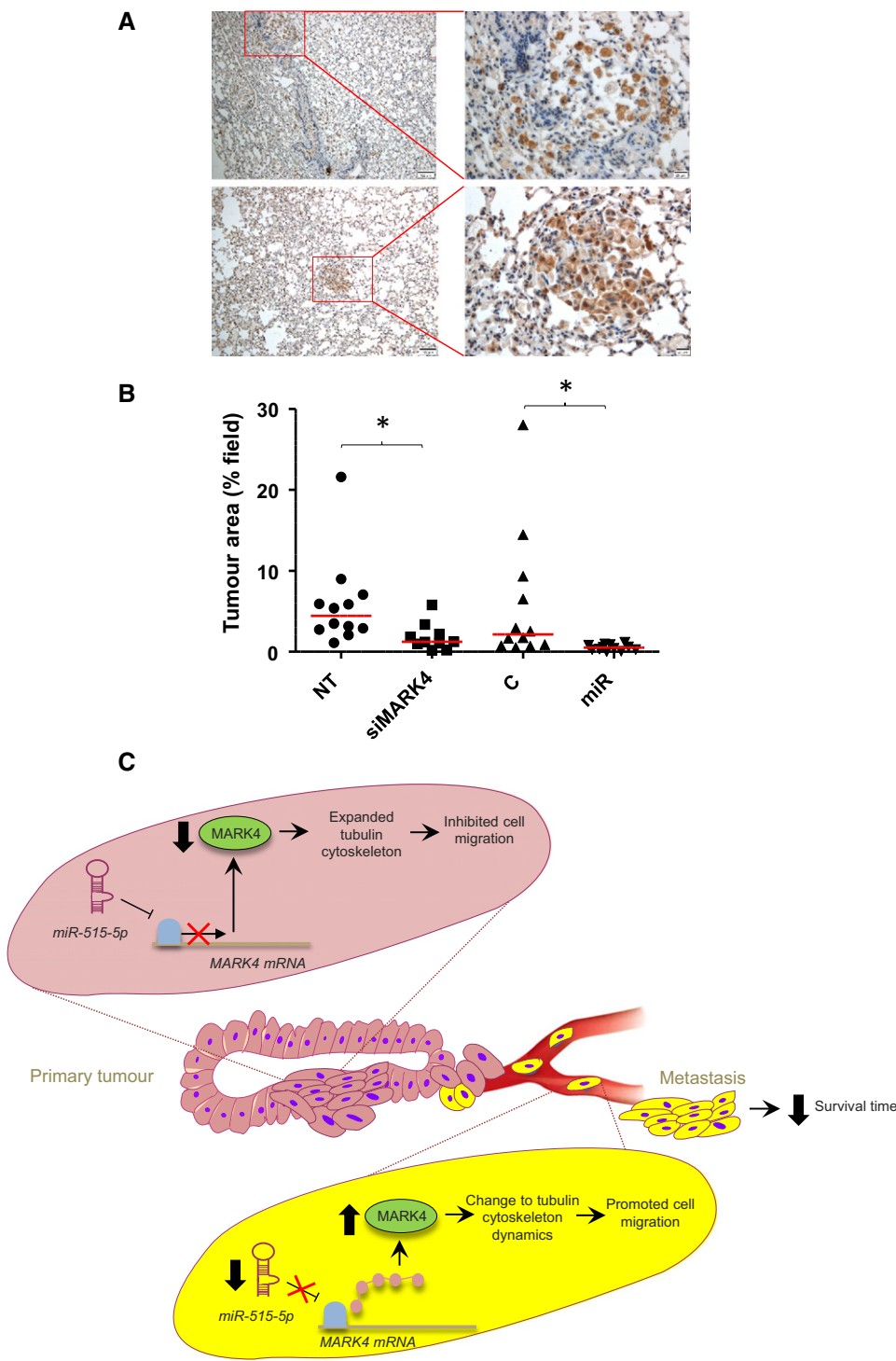

**Figure 7.  miR-515-5p overexpression or siRNA-mediated MARK4 silencing prevents A549 cell metastasis to the lung.**

A549-Luc cells transfected either with miR-515-5p precursor (miR), siRNAs to MARK4 (siMARK4) or the corresponding control sequences (C and NT, respectively) were injected in SCID mice through the tail vein. Four weeks later, the mice were sacrificed and their lungs extracted and fixed prior to staining for luciferase expression.

A  Representative pictures showing the presence of A549-Luc cells in the lungs of NT mice either as single cells (upper panel) or as a tumour mass (lower panel). Scale bar: 100 μm.

B  Percentage of the area covered by tumour cells on randomly observed fields of views. Quantification was performed using Photoshop CS4. Each spot corresponds to quantification of an individual 1 mm × 1.5 mm field. Transversal bar: median; Statistics: Student's *t*-test. *$P \leq 0.05$.

C  Schematic summary. Increase in MARK4 expression is the main reason for miR-515-5p-down-regulated cells to show increased metastatic abilities in breast and lung cancer. The down-regulation of miR-515-5p de-represses MARK4 translation in individual cells from the primary tumour promoting their metastatic dissemination through the blood or lymphatic circulation and reducing patients' survival.

validated this using luciferase report assays that demonstrated that miR-515-5p interacts directly with the 3′ UTR of NRAS, PIK3C2B and MARK4 (Fig 2E and F) and showed the same targets to be down-regulated by miR-515-5p in NSCLC cells (Fig 2D). We focussed on investigating whether MARK4 down-regulation mediated the migratory effects of miR-515-5p because this target was most dramatically decreased by miR-515-5p overexpression in breast cancer (Fig 2B and C) and had previously been involved in micro-tubule dynamics [6]. TargetScan predicted two different sites of miR-515-5p interaction in MARK4 3′ UTR and using site directed mutagenesis coupled with luciferase assay and overexpression of miR-515-5p mimic we demonstrated that miR-515-5p post-transcriptionally repress MARK4 expression interacting with both sites (Appendix Fig S5). Interestingly, MARKs phosphorylate micro-tubule-associated proteins (MAPs) on their microtubule-binding domain, disrupting their binding to tubulin and the dynamics of microtubules. In agreement with our hypothesis, silencing MARK4 mimics the changes in the tubulin cytoskeleton and inhibition of cell migration obtained downstream of miR-515-5p overexpression in all cell lines tested (Fig 3A–C and Appendix Fig S6C). This correlated with an increase in the rate of microtubule polymerisation downstream of MARK4 silencing (Appendix Fig S9). MARK activity has been shown to be regulated by a number of different processes including phosphorylation, ubiquitination and interaction with inhibitory proteins [20]. However, to our knowledge, this is the first report of the regulation of a MARK family member by a micro-RNA.

To test our hypothesis that MARK4 down-regulation was the mechanism through which miR-515-5p affected breast cancer cell migration, we tested whether overexpressing MARK4 in miR-515-5p-transfected cells could rescue the reduced cell motility. In agreement with our hypothesis, MARK4 overexpression rescued the effect of miR-515-5p on random cell migration (Fig 4A and B). However, increased MARK4 expression did not completely rescue the effect of miR-515-5p overexpression on directional migration in two out of the three cell lines tested (Fig 5A and B). This may be due to the additional factors regulating directional cell migration, such as sensing of growth factor gradient and cell deformability that may require input from additional miR-515-5p targets. However, since rescue of both types of migration was almost total (80–100%), our data suggest that MARK4 down-regulation is probably the main mediator of cell migration regulation by miR-515-5p.

We wished to assess whether these *in vitro* finding showed any clinical relevance and quantified miR-515-5p in tissues from breast cancer patients. We found that miR-515-5p expression was significantly reduced in lymph nodes metastasis compared to primary tumours derived from the same breast cancer patients (Fig 6E and Appendix Fig S12). Moreover, we found that low miR-515-5p expression correlated with poorer survival in breast and lung cancer (Fig 6A–D). The opposite correlation existed for MARK4 expression, with high expression of this kinase correlating with poorer survival in these two cancer types (Fig 5A and B). Moreover, MARK4 was clearly up-regulated during lung tumorigenesis and further increased in metastatic lesions in over 40% of cases (Fig 5C–F). MARK4 has previously been shown to be up-regulated in glioma and hepatocellular carcinomas [1,2] and associated with Wnt-induced tumorigenesis in the prostate [21]. Nevertheless, this is the first report of its overexpression during lung tumorigenesis and of its particular association with the metastatic process. This link

between changes in MARK4 expression and clinical metastatic disease was supported by our experimental data using animal models. Indeed, we quantified MARK4 levels in metastatic lesions obtained following the inoculation of MDA-MB-231 cells into the mammary fat pads of nude mice. MARK4 mRNA levels, but not those for NRAS and PI3KC2B, were increased in metastatic as compared to primary lesions in these animals (Fig 6F and Appendix Fig S13). Interestingly, we confirmed the opposite correlation for miR-515-5p with the levels of this miRNA being down-regulated in metastatic lesions (Fig 6F). These data reinforce the notion that MARK4 is the main miR-515-5p target regulating cell migration and metastasis. However, while these data suggested that the miR-515-5p/MARK4 axis may be regulated in the context of metastatic disease, it did not draw a direct link between miR-515-5p expression levels and the onset of metastasis. Hence, we performed an experiment to assess how overexpression of this miRNA would influence to ability of cancer cells to colonise SCID mice following tail vein injection. Contrary to cells transfected with a non-targeting control, miR-515-5p-overexpressing A549 cells failed to establish tumours in these animals (Fig 6G) demonstrating the direct ability of miR-515-5p to regulate the metastatic abilities of cancer cells. In addition, similarly to miR-515-5p overexpression, MARK4 silencing prevented lung colonisation by A549 cells (Fig 7A and B), showing that the reduction of invasive potential observed *in vitro* following MARK4 down-regulation extends to the *in vivo* setting.

In summary, we demonstrate here that miR-515-5p overexpression inhibits cell migration by down-regulating MARK4 mRNA levels in both breast and lung cancer (Fig 7C), suggesting the importance of miR-515-5p/MARK4 mechanism across different cancer types. We showed that miR-515-5p and MARK4 levels are significantly down-regulated/up-regulated in metastatic breast cancer tissues compared to primary tumour tissues, respectively, demonstrating the clinical relevance of our findings and indicating in the future that perhaps a miR-515-5p-based therapy as possible treatment to prevent the migration of cancer cells from the primary tumour. Finally, the correlation between miR-515-5p/MARK4 expression and the outcome of metastatic breast and lung cancer suggests the use of miR-515-5p and MARK4 as prognostic molecular biomarkers for metastatic cancer patients. Also, as MARK4-specific inhibitors have now been developed [21], our data suggest that these molecules may be useful in the clinic to prevent metastatic dissemination of lung, breast and possibly other cancers.

## Materials and Methods

### Cell lines

The human MCF7, MDA-MB-231, MDA-157, A549 and H1299 were obtained from ATTC (Manassas, VA, USA) and maintained in Dulbecco's modified Eagle's medium (Gibco) supplemented with 10% foetal bovine serum (Gibco), 100 U/ml penicillin and 100 mg/ml streptomycin at 37°C with 5% $CO_2$.

### Luciferase assay

MCF7 and MDA-MB-231 cells were plated in 24-well plates and were allowed to adhere and grow overnight at 37°C with 5% $CO_2$. The

following day, the cells were co-transfected with pLightSwitch_3UTR GoClone vectors, and the indicated 50 nM of miRNA precursors (pre-miR mimics, Ambion) using Lipofectamine™ 2000 (Invitrogen). The sequences for the 3′ UTRs used were those for the following Gene IDs: PIK3C2B Gene ID: 5287; NRAS Gene ID: 4893; MARK4 Gene ID: 57787. After 24 h, cells were lysed using a passive lysis buffer (Promega, Southampton, UK) and processed with the LightSwitch Assay System (Switchgear Genomics Menlo Park CA, USA) according to manufacturer's instructions. Luciferase activity detection was performed using a GLOMAX 96 Microplate luminometer (Promega).

### MicroRNAs, siRNAs and plasmids transfection

Cells were plated and allowed to adhere and grow overnight at 37°C with 5% $CO_2$. The following day, cells were transfected with 50 nM of pre-miRs (Ambion) or 30 nM of siRNAs (Dharmacon, Appendix Table S1) using the HiPerFect transfection reagent (Qiagen) according to the manufacturer's instructions. Twenty-four hours later, cells were transfected with control (pCDNA 3.0) or HA-MARK4 plasmid (kind gift from Dr. Drewes [6]) using Attractene and according to the manufacturer instructions.

### Immunofluorescence

Cells were fixed in 4% PFA at 37°C for 15 min, washed in PBS and then blocked in 3% BSA at room temperature for 1 h. Actin cytoskeleton was stained with Alexa 488 Phalloidin (Invitrogen), microtubules with an anti-beta-tubulin antibody (Sigma) and nuclear DNA visualised using DAPI (Invitrogen) at 4°C overnight. The following day, the stained cells were washed twice with PBS, secondary TRITC-antibody added for 1 h and the cells washed thrice with PBS. Images were acquired using an ImageXpress High Content Screening microscope (Molecular Devices), 48 fields per condition.

### Cellular circularity factor

Actin cytoskeleton staining of miR-515-5p and control miRVANA transfected cells was performed using Alexa 488 Phalloidin as described above, and fluorescent microscopy images for 10 fields per condition were acquired on an ImageXpress screening microscope (Molecular Devices). The cellular circularity factor was calculated for individual cells in ImageJ following the formula $4\pi A/P2$, where A is the cell area and P is the cell perimeter.

### Cell cycle analysis by FACS

The Click-iT® EdU cell proliferation assay kit (Life Technologies) was used to assess changes in cell cycle profile. Following the desired treatment, before harvesting the cells, 5-ethynyl-2′-deoxyuridine (EdU) was added to the culture medium (10 μM) for 1 h, at 37°C 10% $CO_2$, to allow for incorporation into the DNA during active DNA synthesis. Cells were then trypsinised, pelleted and washed with 1% (w/v) BSA in PBS. About 100 μl of Click-iT® fixative was added and incubated at room temperature, in the dark, for 15 min. The cells were again washed with 1% BSA PBS and permeabilised with 100 μl of 1× Click-iT® saponin-based permeabilisation buffer/wash reagent for 15 min. As per the manufacturer's instructions, 1× Click-iT® reaction cocktail was prepared (for 1 ml: 100 μl 1× Click-iT® EdU

buffer additive, 875 μl PBS, 20 μl $CuSO_4$, 5 μl Alexa Fluor® 488 azide) and 500 μl added to the cells at room temperature for 30 min in the dark. Cell pellets were then washed with 1× Click-iT® saponin-based permeabilisation and wash reagent once before being re-suspended in 500 μl 1× Click-iT® saponin-based permeabilisation and wash reagent. The stained cells were analysed using the FACS-Diva software v6.12 on a FACSCanto A flow cytometer. Data were analysed using FlowJo v7.6.5 (Tree Star Inc).

### Cell tracking assay

$1 \times 10^3$ cells per well were plated in black, flat and clear bottomed 96-well plates (BD Biosciences). Time-lapse imaging was performed for 18 h (1 image/10 min) using a motorised-staged environment-controlled ImageXpress high content screening microscope powered by MetaXpress 2.0 (Molecular Devices). Two sites were acquired per well, and each condition was present in triplicate. To quantify the degree of migration, a minimum of 30 cells per condition were tracked using MetaXpress Track Points application. Tracks were analysed using a previously published Mathematica (Wolfram Research) notebook [22].

### Boyden chamber assay

For transwell migration assays, $5 \times 10^4$ cells were seeded at the top uncoated membranes with 8.0-μm pores (BD Biosciences). Cells were plated in serum-free medium and allowed to migrate towards a complete growth medium for 8 h. The cells on the bottom part of the filter were then stained using crystal violet. Stained cells were then imaged using Olympus BX51 microscope. Directional migration was quantified by cell counting using ImageJ [23].

### Microtubule comet imaging and analysis

To analyse the speed of microtubule growth, a plasmid encoding GFP-tagged end-binding protein 3 (EB3) was used. The plasmid was transfected into A549 cells 24 h before the cells were re-plated into 8-well chamber slides (PAA laboratories). During re-plating, cells were reverse transfected with siRNA against the MARK4 or a non-targeting control siRNA. Cells were then incubated for 24 h at 37°C 10% $CO_2$. The movement of the EB3-GFP comets was imaged in individual cells using a Zeiss LSM-780 inverted microscope (Zeiss) using the Zen Black software (Zeiss) and a 40× magnification lens with one image acquired per sec for a total of 2 min on a 1.3-μm z-section. A total of 12 cells per condition were imaged in each independent repeat experiment. Comets were tracked using FIJI ImageJ (NIH, USA) and the plug-in wrMTrck (J. Pedersen) to determine comets' speed.

### RNA-Seq and ChIP-Seq analysis

Two micrograms of total RNA from each sample was used to produce cDNA libraries from polyA enriched RNA using the True-Seq RNA preparation kit (Illumina Inc., San Diego, CA, USA) according to the manufacturer instructions. Paired end sequences (reads) 100 nt in length were then generated using a HiSeq 2000 instrument (Illumina). Fastq files containing the sequenced reads, obtained at the end of the sequencing, were mapped to the University of California at Santa Cruz (UCSC) human genome (hg19 assembly) with TopHat version 1.4.1 (http://tophat.cbcb.umd.edu),

using default settings. The mapped bam files obtained at the end of the runs were then analysed using Cufflinks version 2.1.0 (http://cufflinks.cbcb.umd.edu) for RNA quantification and analysis. The RNA-seq raw data have been deposited in the ENA database with accession number PRJEB11884.

### MicroRNAs and mRNA expression

RNA was isolated using TRIzol reagent according to the manufacturer instructions. MicroRNA levels were quantified using TaqMan MicroRNA Reverse Transcription kit (Applied Biosystems), TaqMan 2× Universal PCR Master Mix No AmpErase UNG (Applied Biosystems) and has-pre-miR-515-5p TaqMan primers (Applied Biosystems). To analyse mRNA expression, both cDNA conversion and qPCR amplification were performed as recommended by Applied Biosystems using the High-Capacity cDNA Reverse Transcription Kit and Fast SYBR® Green Master Mix, respectively. The primers used for NRAS, PI3KC2B, FZD4, CDC42BPA, MARK4 and GAPDH are detailed in Appendix Table S2.

### Tumour tissues—microRNA expression

MicroRNA expression analysis in formalin-fixed paraffin-embedded (FFPE) was approved by a UK national research ethics committee (London; 07/Q0401/20) and by Imperial College Healthcare NHS Trust. We prospectively measured miR-515-5p expression in primary BCs and corresponding lymph node metastasis from patients with lymph node positive BC. Between January 2008 and February 2010, we obtained primary breast tumours and corresponding lymph node metastases at Imperial College Healthcare NHS Trust, London, UK. All samples were formalin-fixed and paraffin-embedded. Clinico-pathological information about the patients is provided in Appendix Table S3.

### Lung cancer versus normal lung cDNA arrays

The levels for MARK4 mRNA were determined in the HLRT104 TissueScan cDNA array (Origene) containing 24 lung cancer and matching normal lung samples and were analysed by using the Fast SYBR Green PCR Master Mix and 7900HT Real-time PCR System (Applied Biosystems). The housekeeping genes hypoxanthine phosphoribosyl-transferase (HPRT) and glyceraldehyde-3-phosphate dehydrogenase (GAPDH) were used as reference genes to normalise the expression of target genes. Primers used are presented in Appendix Table S2. Analysis of relative gene expression was carried out using the qbasePLUS 2.1 software (Biogazelle).

### Immunohistochemistry

The lung cancer tissue microarray (TMA) BC041114 containing 90 cases of lung carcinoma and 10 normal tissue samples was purchased from Insight Biotechnology Limited (UK). For the syngeneic primary tumour/metastasis tissue microarray, 100 lung cancer cases (Ethic Ref: 06/Q0406/154) were considered suitable on the basis of histotype confirmation and immunostaining for pan-cytokeratin MNF116 and vimentin. Tissue microarrays were prepared by obtaining three 1-mm cores from the most representative and better preserved areas of the tumours and re-embedded in microarray blocks.

TMAs were stained using a primary antibody against MARK4 (Sigma-Aldrich, SAB1300394) used at a 1:50 dilution. A semiquantitative scoring of the histochemical staining was used as previously published by us [24]. The intensity of the staining was given a score of 0, negative, to 3, strongly positive. The percentage coverage of the staining was assessed and scored between 0 and 100%. Scores were multiplied together to give a range of possible scores between 0 and 300. The cores were all scored bind of all clinical data. The TMA data for each array were analysed separately then appropriately grouped to assess any changes in MARK4 protein levels in tissue within different stage, grade and classification of tumours.

### Animal experiment

A549 cells stably expressing luciferase (A549-Luc-C8, Xenogen) were transfected with 50 nM pre-miR-515-5p or miRVANA control using HiPerFect (Qiagen) or with 20 nM MARK4 or non-targeting siRNAs. Forty-eight hours later, $1 \times 10^6$ cells were injected in the tail vein of SCID mice (eight animals per group) and mice subjected to whole-body imaging under anaesthesia, at 1 and 4 weeks, following tail vein administration of luciferin or sacrificed at 4 weeks prior to their lungs being extracted, fixed in 70% ethanol, paraffin-embedded and stained for luciferase expression by immunohistochemistry (anti-luciferase antibody, sc-32896 from Santa Cruz Biotechnology). For the latter, images were acquired under microscopic observation and at ×10 and ×40 magnification. The percentage of field of view area covered by luciferase staining in 10× magnification images was quantified under FIJI-ImageJ. Animal experiments were carried out according to the National Guidelines for Experimental Animal Welfare (MOST, P.R. China, 2006) in a SPF facility certified by the Hubei Provincial Office of Science and Technology (SYXKE 2009−0051).

### Statistical analysis

All data in the figures are presented as the mean ± standard error of mean (SEM) and assessed by using one-way ANOVA analysis followed by a Tukey's test (95% confidence) for multiple comparisons (SPSS version 17). *P*-value of < 0.05 is considered statistically significant.

**Expanded View** for this article is available online.

### Acknowledgements

We thank Harikrishna Nakshatri for the MDA-MB-231 cells derived from metastatic loci in nude mice. We would like to thank Builder Depot Ltd and The Harold and Daphne Cooper Charitable Trust, the Fundacão para a Ciência e Tecnologia, Portugal (FGP; SFRH/BD/68267/2010), and the Medical Research Council with Johnson & Johnson (CEM, MRC CASE G1000390 and JK) for their support. Also OP was supported by EU FP7 funding under the LungTarget consortium. In addition, this study was supported by the Imperial Experimental Cancer Medicine Centre (ECMC), the NIHR Imperial Biomedical Research Centre (BRC), Cancer Treatment and Research Trust (CTRT) and the Association for International Cancer Research (AICR). For additional support, we acknowledge Action Against Cancer, the NIHR (JS), Cancer Research UK (JS and RL), the Association for International Cancer (LC), the Breast Cancer Campaign (LP) and the Roter Family Foundation. This article is written in memory of Leslie Aaronson.

## Author contributions

JS, CEM, FGP, LC and MJS involved in conception and design; CEM, FGP, JK, OEP, RL and LP involved in the development of methodology; OEP, LC, CEM and FGP acquired the data; OP, CEM, FGP, LC and JS analysed and interpreted the data; OEP, CEM, FGP, JS, LC, JW and MJS wrote, reviewed and revised the manuscript; JS, MJS, and JW provided administrative, technical or material support; OEP, JS, LC, MJS, JW and DP supervised the study. YH, FM and AEF performed experiments. YW supervised the study.

## Conflict of interest

The authors declare that they have no conflict of interest.

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
