## [Review Process File · EMBO Reports]

Manuscript EMBO-2015-40970

miR-515-5p controls cancer cell migration through MARK4 regulation

Catriona Munro, Leandro Castellano, Ms. Yili Hu, Francesco Mauri, Jonathan Krell, Romain Lara, Filipa Pinho, Loredana Pellegrino, Dmitry Pshezhetskiy, Yulan Wang, Jonathan Waxman, Michael Seckl, Justin Stebbing and Olivier Pardo

Corresponding author: Olivier Pardo, Imperial College

Review timeline:

Submission date:	03 July 2015
Editorial Decision:	05 August 2015
Revision received:	23 November 2015
Editorial Decision:	16 December 2015
Revision received:	22 December 2015
Accepted:	13 January 2016

Transaction Report:

1st Editorial Decision

05 August 2015

Thank you for your submission to EMBO reports. We have now received the two enclosed reports on it. As you will see, although the referees find the topic of interest, they both raise numerous serious concerns and are ultimately skeptical of the physiological relevance of your data, for which they believe you have provided insufficient support.

As the reports are below, I will not detail them here. However, it is clear that substantial improvement of the statistical analyses throughout the study, and considerable strengthening of the *in vivo* experiments and clinical validation, as well as a direct demonstration of the importance of MARK4 in metastasis *in vivo*, are needed for further consideration in EMBO reports. All referee concerns are pertinent, and addressing them would be a pre-requisite for publication in EMBO reports.

REFeree REPORTS

Referee #1:

In this manuscript, Pardo et al aim to address the role of microRNA miR515-5p in cancer progression and metastasis. Using cell culture, the authors analyze the role of miR515-5p in migration and identify MARK4 as a target gene and a functionally relevant mediator of the process. They use gene expression profiles and tissue microarray to analyze clinical relevance and mouse

models for functional analysis. This is an interesting study, where the link between miR515-5p and MARK4 in cell lines and their role in migration are generally conclusive. However, there are significant concerns with regard to the *in vivo* experiments and clinical validation that need addressing.

Specific points.

1. A link between miR515-5p and MARK4 in cancer patient material is missing. The investigators address clinical relevance of miR515-5p and MARK4 expression in tumors using Kaplan Meier analysis of patient-datasets (figures 5a,b and 6a,b). The concordance between the prognostic value of MARK4 and miR515-5p is rather weak in these patient samples. In lung adenocarcinoma, MARK4 associates with poor overall survival while ER-neg breast cancer shows only a modest trend. On the other hand, low miR515-5p expression is strongly linked to poor outcome in ER-neg breast cancer but not convincingly in lung cancer. Notably, the patient cohorts used to analyze MARK4 and miR515-5p are distinct and therefore, confounding effects cannot be excluded. Still the lack of robust concordance makes it hard to draw conclusions. Correlative evidence could be derived by analyzing MARK4 expression in primary tumors and metastases (already analyzed for miR515-5p in figure 6c).
2. In figure 5c-f, MARK4 expression is analyzed in human cancer and normal tissue samples. It is not clear to this reviewer how to interpret these results. According to figure 5 c-f, certain tumors show an increase in MARK4 expression while others show a decrease. But without clinical parameters it is hard to conclude anything from this. Is there anything in particular associated with MARK4 expressing or MARK4 negative cancers? Moreover, it would be interesting to see if miR515-5p expression correlated negatively with MARK4 in any of these samples.
3. Functional analysis of miR515-5p *in vivo* needs improvement. The xenograft model for lung cancer, shown in figure 6e and 6f, does not look right. Based on the bioluminescence, the cancer cells did not reach the lungs after tail vein injection. Moreover, quantification does not seem to be normalized to bioluminescence at day 0. This is important to clarify. Finally, the functional role of miR515-5p in breast cancer metastasis should also be addressed, for example by using MDA-MB-231 cells which efficiently metastasize from the mammary gland.
4. *In vitro* experiments, linking miR515-5p and MARK4 to motility, are generally conclusive. However, regulation of microtubule dynamics would also be expected to have significant effects on cell division. This should be tested.
5. Throughout the manuscript, the authors analyze expression of MARK4 but do not address MARK4 activity or its association with microtubules.
6. The authors state that miR515-5p expression levels were significantly lower in MDA-MB-231 compared to MCF7 (page 10). However, in their previous paper, they report that the ER signaling suppresses miR515-5p expression and show that ER negative human breast cancer expresses lower miR515-5p levels compared to ER positive cancer (Pinho et al 2013). This should be discussed.
7. miR515-5p induced effect on polarity is mentioned in several places within the manuscript, but is not rigorously analyzed.
8. To confirm that MARK4 is a direct target of miR515-5p, it is important to mutate the seed target site in the 3'UTR that was used in the reporter assay.

Referee #2:

In this manuscript, the role of the micro RNA miR-515-5p in cancer cell migration has been investigated. By over-expressing miR-515-5p in two breast cancer cell lines and analyzing changes in gene expression by RNAseq, 5 potential targets were identified including the MARK4 protein kinase. Direct knockdown of MARK4 led to similar effects on altering cell morphology, microtubule reorganization and inhibiting cancer cell migration at miR-515-5p over-expression,

while MARK4 over-expression reversed the inhibitory effects of miR-515-5p, consistent with MARK4 being the important target for these cell responses. Analysis of miR-515-5p and MARK4 expression in human and mouse tumors indicated that reduced miR-515-5p levels and/or elevated MARK levels are associated with reduced survival and/or increased metastasis, suggesting a causal relationship between the transcriptional regulation of these RNA transcripts and tumor progression. In vivo experiments with transplanted MDA MB 231 cells indicated that miR-515-5p had a significant role in determining metastatic potential using two models of breast cancer metastasis. These observations would be of interest to cancer and microRNA researchers.

The manuscript is clearly written and details the rationale, findings and conclusions of the study effectively.

Statistical analysis is a major issue with this manuscript, and which drastically detracts from the robustness of the conclusions. In every instance, individual replicates for control groups have been normalized to 100% values a priori, such that the mean values are 100% +/- 0. This approach is absolutely incorrect as it artificially suppresses the true variance in the control groups, which invalidates their appropriateness for statistical analysis. Why this has been done for assays that generate quantifiable absolute measurements (e.g. distance travelled in Figure 1B, percentage migratory cells in Figure 1C, tumor burden in Figure 6E, etc.) is perplexing. In every instance, this should be changed so that absolute values are used whenever possible. In those instances where relative values or arbitrary units are being compared, such as the fold change in mRNA levels in Figure 2, the mean values for the control condition should be determined post hoc, and then used for normalization of the individual control and experimental data points. This would enable legitimate statistical analysis of control versus experimental conditions.

Additional problems with data presentation are the absence of units in graphs such as cell migration in Figure 1B, and the absence of scale bars in micrographs such as in Figure 1C. These deficiencies should be corrected throughout the manuscript.

Analysis of publicly available gene expression data was used to compare miR-515-5p levels in metastatic versus non-metastatic lung cancer patients on page 13, but the data is not shown. Given that the authors have done this comparison and concluded that it is not significant, it would be important to either show the data (supplemental would be satisfactory) or not report this insignificant observation.

The in vivo experiments support the conclusion that miR-515-5p has a major effect on the metastatic ability of MDA MB 231 cells to metastasize in the two models used, but only show an association with MARK4 expression that does not support a causal role. In order to fully support the conclusions of the manuscript, it would be necessary to show a direct role for MARK4 in these metastatic behaviours, which could be showing that MARK4 knockdown reduced metastasis, either on its own or in combination with miR-515-5p knockdown.

The supplementary data provided is excessive and often unnecessary or so incompletely described as to be essentially useless. Supplementary Table S2 only needs the reference cited in the Methods section. Supplementary Table S5 would be considerably more useful if information were provided on fold-change and statistical significance. In addition, it would be helpful if there were a direct comparison between the experimental data on transcripts regulated by miR-515-5p over-expression in both MCF7 and MDA MB 231 cells, and the predictions made by Target scan in silico analysis. Although the disclosure of these data is to be commended, their utility for the research community are limited by the absence of accompanying information. The coding sequences of genes in Supplementary Table S6 are unnecessary, it would be sufficient to cite NCBI reference numbers for each gene in the Methods section.

1st Revision - authors' response

23 November 2015

Thank you very much indeed for your helpful remarks and those of the referees. We have enormously improved the manuscript now and re-written large sections of it having performed major additional experiments. A reasonable proportion of these new data were not requested, such

as the comet tunnel assay to look at cell migration speed, but strengthen the paper enormously. Overall the manuscript we re-submit is much improved.

In particular, we have now performed an additional animal experiment to answer one of the comments from Reviewer #2 and strengthen the link between miR-515-5p and MARK4 in view of comments from Reviewer #1.

In short, we believe that we have answered all the reviewers' comments satisfactorily and trust this makes our manuscript acceptable now for publication in EMBO Reports.

Corrections to the text in the new improved version of the manuscript have been inserted in red so as to facilitate the task of the reviewers in assessing our additions.

In answer to the specific reviewer comments:

Referee #1:

In this manuscript, Pardo et al aim to address the role of microRNA miR515-5p in cancer progression and metastasis. Using cell culture, the authors analyze the role of miR515-5p in migration and identify MARK4 as a target gene and a functionally relevant mediator of the process. They use gene expression profiles and tissue microarray to analyze clinical relevance and mouse models for functional analysis. This is an interesting study, where the link between miR515-5p and MARK4 in cell lines and their role in migration are generally conclusive. However, there are significant concerns with regard to the in vivo experiments and clinical validation that need addressing.

Specific points.

1. A link between miR515-5p and MARK4 in cancer patient material is missing. The investigators address clinical relevance of miR515-5p and MARK4 expression in tumors using Kaplan Meier analysis of patient-datasets (figures 5a,b and 6a,b). The concordance between the prognostic value of MARK4 and miR515-5p is rather weak in these patient samples. In lung adenocarcinoma, MARK4 associates with poor overall survival while ER-neg breast cancer shows only a modest trend. On the other hand, low miR515-5p expression is strongly linked to poor outcome in ER-neg breast cancer but not convincingly in lung cancer. Notably, the patient cohorts used to analyze MARK4 and miR515-5p are distinct and therefore, confounding effects cannot be excluded. Still the lack of robust concordance makes it hard to draw conclusions. Correlative evidence could be derived by analyzing MARK4 expression in primary tumors and metastases (already analyzed for miR515-5p in figure 6c).

We thank the referee for making this clever point. We agree with the referee that the concordance between the prognostic value of MARK4 and miR515-5p is rather weak in these patient samples and as the referee pointed out, this could be due to the fact that the analysis for miR-515-5p and MARK4 was done in independent cancer samples. Unfortunately, we cannot measure the levels of MARK4 mRNA in the samples used to measure miR-515-5p because those samples were rather small and no material is available anymore for further analysis.

However, to address the reviewer's comment, we now have analysed large cohorts of clinical samples from publicly available databases and were able to show inverse correlation between miR515-5p and MARK4 expression in breast cancer.

As the referee pointed out low expression of miR515 in our lung cancer set is not convincing (Figure 6B). In order to further demonstrate the clinical relevance of this miRNA we now, in aggregate analysed:

*195 samples of lung adenocarcinoma from the TCGA study;
34 samples of lung squamous cell carcinoma from the TCGA;
60 sample of breast cancer from Lyng Ditzel Breast study (GSE37405);
44 samples of breast cancer from the GSE37405.*

In all these cases, low miR-515 expression was linked to poor survival and all the Kaplan Meier plots showing these results are now depicted in Supplemental Figure 10. Reference to these new results has been added on Page 14 of the manuscript.

2. In figure 5c-f, MARK4 expression is analyzed in human cancer and normal tissue samples. It is not clear to this reviewer how to interpret these results. According to figure 5 c-f, certain tumors show an increase in MARK4 expression while others show a decrease. But without clinical parameters it is hard to conclude anything from this. Is there anything in particular associated with MARK4 expressing or MARK4 negative cancers? Moreover, it would be interesting to see if miR515-5p expression correlated negatively with MARK4 in any of these samples.

We agree with the reviewer that having data on the metastatic status of the patients associated with the analysed samples would have provided additional information on the clinical relevance of MARK4 expression in disease progression. Unfortunately, the clinical data available for the analysed samples did not contain information about the extent of tumour dissemination and it is therefore impossible for us to comment on this.

As for the correlation between MARK4 and miR515-5p levels, unfortunately, the mentioned samples have been completely used and are not available anymore.

However, as pointed in response to the previous question from this reviewer, we re-analysed TCGA level 3 data of normalised values and found that the miR-515-3p is negatively correlated with MARK4 in some breast cancer sub-types. To further demonstrate correlation between the two molecules in ex vivo samples we applied Pearson correlation on the expression values of miR-515-5p and MARK4 and found significant negative correlation at least in luminal B and normal counterparts. This new information is now added in Supplementary figure 11. Text describing these new data has been added on Page 14 of the manuscript.

3. Functional analysis of miR515-5p in vivo needs improvement. The xenograft model for lung cancer, shown in figure 6e and 6f, does not look right. Based on the bioluminescence, the cancer cells did not reach the lungs after tail vein injection. Moreover, quantification does not seem to be normalized to bioluminescence at day 0. This is important to clarify. Finally, the functional role of miR515-5p in breast cancer metastasis should also be addressed, for example by using MDA-MB-231 cells which efficiently metastasize from the mammary gland.

The normalisation of imaging signal intensity was not done to Day 0 because this process squashes the data due to the high total signal obtained from the injected cells, making plotted results unreadable. Indeed, the amount of cells surviving the initial 4 days following injection is minimal due to cell death occurring in circulation and to normalise the data at 4 weeks to the initial amount of surviving cells at Day 4 is therefore more appropriate.

In addition, to answer the reviewer's concern that the injected cells did not reach the lungs of the animals, we present new data in Fig 7A showing that A549 adenocarcinoma cells reach the lungs of the animals and develop into tumours 4 weeks following injection into Scid mice. However, as shown by the bioluminescence imaging (Fig 6E), other sites are also efficiently (if not better) colonised by these cells.

Unfortunately, we were unable to perform the same miR515-5p luminescence imaging-based experiments using MDA-MB-231 cells as we did not have access to the corresponding luciferase-expressing cell line. Instead, to extend our findings to the regulation of metastasis by MARK4, we performed an additional animal experiment in Scid mice with tail-vein injection of MARK4-silenced A549-Luc cells, or corresponding control cells. These experiments show that MARK4 silencing prevents lung colonisation by A549 cells and results to that effect are presented on Fig 7B with corresponding text on Page 15.

4. In vitro experiments, linking miR515-5p and MARK4 to motility, are generally conclusive. However, regulation of microtubule dynamics would also be expected to have significant effects on cell division. This should be tested.

We thank the referee for this suggestion and, to address this issue, performed additional experiments which are now presented as part of the manuscripts. We compared the cell cycle distribution between control-transfected A549 cells and cells silenced for MARK4. In concordance with the referee's prediction, this analysis revealed that MARK4-downregulated cells accumulated in the G1 phase of the cell cycle while showing a decrease in DNA synthesis. In addition we found that cell cycle arrest exerted by MARK4 silencing was accompanied by an increase in the levels of

p53 as well as its transcriptional target, p21, in MARK4-silenced cells. These results has been added in Supplementary Figure 8 and corresponding text inserted on Page 12.

5. Throughout the manuscript, the authors analyze expression of MARK4 but do not address MARK4 activity or its association with microtubules.

We have now performed the requested experiments. To analyse how MARK4 impacts on the speed of microtubule growth, a plasmid encoding GFP tagged End Binding Protein 3 (EB3) was transfected into A549 cells. EB3 binds to the growing end of the microtubules (MT) and by imaging the movement of the GFP comets produced as the microtubule grows we were able to quantify the speed of MT growth. We compared this growth rate between cells silenced for MARK4 or transfected with a non-targeting control siRNA. Comet speed was found to be significantly higher in cells silenced for MARK4 suggesting an increase in the rate of tubulin polymerisation. This is shown in Supplementary figure 9 and the corresponding text in inserted on Page 13 of the manuscript.

6. The authors state that miR515-5p expression levels were significantly lower in MDA-MB-231 compared to MCF7 (page 10). However, in their previous paper, they report that the ER signaling suppresses miR515-5p expression and show that ER negative human breast cancer expresses lower miR515-5p levels compared to ER positive cancer (Pinho et al 2013). This should be discussed.

Although MDA-MB-231 cells do not express ERs that, as shown in our previous manuscript, suppresses miR-515-5p levels, we think that this miRNA may also be regulated by other factors that are linked to cytoskeleton remodelling. These factors would suppress miR-515-5p levels, independently of ERα, in the highly metastatic/mesenchymal MDA-MB-231 cells. This speculation has now been added in the discussion.

7. miR515-5p induced effect on polarity is mentioned in several places within the manuscript, but is not rigorously analyzed.

The reviewer was correct in suggesting that the changes in cell polarity should be quantified here. The loss of cell polarity mediated by miR-515-5p over-expression has now been rigorously investigated by comparing the cell circularity factor of individual cells in 10 fields of view between control and miR-515-5p over-expressing conditions for the four cell lines (Supplementary Figure 2). Overexpression of miR-515-5p significantly increased the circularity factor towards a value of 1, indicating loss of cell elongation. A section in the Material and Methods has been added to explain how the circularity factor is calculated and text covering the new findings inserted on Page 9.

8. To confirm that MARK4 is a direct target of miR515-5p, it is important to mutate the seed target site in the 3'UTR that was used in the reporter assay.

We now performed site directed mutagenesis on both the sites predicted by TargetScan and demonstrated that miR-515-5p regulates MARK4 by interacting with both. Indeed, mutation of both sites in the 3'UTR prevents miR-515-5p from inhibiting luciferase expression from our reporter vector. These data are now presented in Supplementary figure 5 and described in the text of the manuscript on Page 11.

Referee #2:

In this manuscript, the role of the micro RNA miR-515-5p in cancer cell migration has been investigated. By over-expressing miR-515-5p in two breast cancer cell lines and analyzing changes in gene expression by RNAseq, 5 potential targets were identified including the MARK4 protein kinase. Direct knockdown of MARK4 led to similar effects on altering cell morphology, microtubule reorganization and inhibiting cancer cell migration at miR-515-5p over-expression, while MARK4 over-expression reversed the inhibitory effects of miR-515-5p, consistent with MARK4 being the important target for these cell responses. Analysis of miR-515-5p and MARK4 expression in human and mouse tumors indicated that reduced miR-515-5p levels and/or elevated MARK levels are associated with reduced survival and/or increased metastasis, suggesting a causal relationship between the transcriptional regulation of these RNA transcripts and tumor progression. In vivo experiments with transplanted MDA MB 231 cells indicated that miR-515-5p had a significant role in determining metastatic potential using two

models of breast cancer metastasis. These observations would be of interest to cancer and microRNA researchers.

The manuscript is clearly written and details the rationale, findings and conclusions of the study effectively.

Statistical analysis is a major issue with this manuscript, and which drastically detracts from the robustness of the conclusions. In every instance, individual replicates for control groups have been normalized to 100% values a priori, such that the mean values are 100% +/- 0. This approach is absolutely incorrect as it artificially suppresses the true variance in the control groups, which invalidates their appropriateness for statistical analysis. Why this has been done for assays that generate quantifiable absolute measurements (e.g. distance travelled in Figure 1B, percentage migratory cells in Figure 1C, tumor burden in Figure 6E, etc.) is perplexing. In every instance, this should be changed so that absolute values are used whenever possible. In those instances where relative values or arbitrary units are being compared, such as the fold change in mRNA levels in Figure 2, the mean values for the control condition should be determined post hoc, and then used for normalization of the individual control and experimental data points. This would enable legitimate statistical analysis of control versus experimental conditions.

We agree with the reviewer that statistical analysis on raw data is in all cases correct. However, we respectfully disagree on the fact that performing statistical analysis on normalised data is incorrect in our particular case.

Indeed, while normalisation would have prevented proper statistical analysis of the data across different cell lines or different sets of experimental conditions (ie. Data obtained with siRNA versus data obtained with miR-515-5p), the process of normalisation does not invalidate pair-wise statistical analysis between samples obtained within the same cell line/experiment (which is the full extent of the statistical analysis presented here for the data discussed by the reviewer). The reason for using data normalisation was to enable clearer representation of the data from different cell lines within a single figure panel. Indeed, the average background distance of cell migration achieved by MDA231 cells, for instance, is higher than that covered by A549 cells (Fig 1C) or the number of H1299 cells transmigrating in control conditions is widely superior to that for A549 cells (Fig 1D). Hence, data normalisation enables here to make the similar trends upon miR-515-5p treatment clearer without confusing the readers with variations in background levels. Furthermore, representing data in this fashion and performing statistical analysis on normalised data appears to be common practice and has been published in various high-impact factor journals, including Embo Reports. Here are few examples of this: Shao-guang Sun et al., EMBO reports (2011) 12, 56-62; Jun He et al., EMBO reports (2012) 13, 1116-1122; Marta Fierro-Fernández et al., EMBO reports (2015) 16, 1358-1377; Lin Zhang et al., Nature (2015) 527, 100-104.

Additional problems with data presentation are the absence of units in graphs such as cell migration in Figure 1B, and the absence of scale bars in micrographs such as in Figure 1C. These deficiencies should be corrected throughout the manuscript.

We thank the reviewer for kindly pointing this out and apologise for these errors. The units for the graphs in Fig1B are micrometers and labels have now been introduced in the figure to reflect this. The missing scale bar has now been introduced in the microscopy pictures of Figure 1C together with the corresponding comment in the figure legend "Scale bar is 20 μm".

Analysis of publicly available gene expression data was used to compare miR-515-5p levels in metastatic versus non-metastatic lung cancer patients on page 13, but the data is not shown. Given that the authors have done this comparison and concluded that it is not significant, it would be important to either show the data (supplemental would be satisfactory) or not report this insignificant observation.

Following the suggestion of the reviewer, as the data showed no statistical significance, the corresponding text has now been deleted from the manuscript on Page 13 and new data from the analysis of publically available databases is now presented instead in Supplementary figure 10 and 11 and corresponding text inserted on Page 14. Indeed, we now have analysed large cohorts of

clinical samples from publicly available databases and were able to show inverse correlation between miR515-5p and MARK4 expression in breast cancer.

As the referee pointed out low expression of miR515 in our lung cancer set is not convincing (Figure 6B). In order to further demonstrate the clinical relevance of this miRNA we now, in aggregate analysed:

*195 samples of lung adenocarcinoma from the TCGA study;
34 samples of lung squamous cell carcinoma from the TCGA;
60 sample of breast cancer from Lyng Ditzel Breast study (GSE37405);
44 samples of breast cancer from the GSE37405.*

In all these cases, low miR-515 expression was linked to poor survival and all the Kaplan meier plots showing these results are now depicted in Supplemental Figure 10.

The in vivo experiments support the conclusion that miR-515-5p has a major effect on the metastatic ability of MDA MB 231 cells to metastasize in the two models used, but only show an association with MARK4 expression that does not support a causal role. In order to fully support the conclusions of the manuscript, it would be necessary to show a direct role for MARK4 in these metastatic behaviours, which could be showing that MARK4 knockdown reduced metastasis, either on its own or in combination with miR-515-5p knockdown.

We have now performed an additional animal experiment to address this comment. A549-luciferase cells were transfected with MARK4 siRNAs or non-targeting controls and injected in Scid mice through the tail vein as in the experiment for miR-515-5p already presented in Figure 6E. Due to our loss of access to a whole-animal imager, in the new experiments, we instead extracted the lungs 4 weeks post-injection, sectioned them following fixation and stained sections with an antibody against luciferase prior to microscopic observation to reveal A549 cells colonisation of the mice lungs. The percentage of surface covered by tumour cells per field of view was then determined by image analysis. A section was added in the Material and Methods of our manuscript to detail this process. In these new experiments, we also injected mice with A549-luciferase cells transfected with miR-515-5p or a non-targeting sequence as an internal positive control in view of results already presented in Figure 6E and to rule out the possibility that our new method may interfere with interpretation of the data. As shown in Figure 7A, miR-515-5p expression still significantly decreased lung colonisation by A549 cells in these new experiments. More interestingly, we now show that MARK4 silencing mimics the effects of miR-515-5p expression and prevents A549 lung metastasis. A corresponding paragraph has been added to the Results section on Page 15.

The supplementary data provided is excessive and often unnecessary or so incompletely described as to be essentially useless. Supplementary Table S2 only needs the reference cited in the Methods section. Supplementary Table S5 would be considerably more useful if information were provided on fold-change and statistical significance. In addition, it would be helpful if there were a direct comparison between the experimental data on transcripts regulated by miR-515-5p over-expression in both MCF7 and MDA MB 231 cells, and the predictions made by Target scan in silico analysis. Although the disclosure of these data is to be commended, their utility for the research community are limited by the absence of accompanying information. The coding sequences of genes in Supplementary Table S6 are unnecessary, it would be sufficient to cite NCBI reference numbers for each gene in the Methods section.

We thank the referee for these suggestions. We now introduced a new Supplemental Table 5 in excel format. We added fold changes and the overlap between the down-regulated genes and TargeTscan prediction as requested. We could not do any statistical analysis since the samples used in RNA-seq were in duplicate but we selected just the genes that were down-regulated in both replicates for further analysis. Supplementary Table S6 has been now removed as requested and the NCBI Gene IDs for each gene cited in the methods section.

Thank you for the submission of your revised manuscript to EMBO reports. We have now received the full set of referee reports that is copied below.

As you will see, both referees are positive about the study and request only minor changes. Referee 1 suggests moving the new clinical data set associating miR-515-5p with prognosis to the main figure as the data set is generated from a larger cohort. Have you also analyzed MARK4 expression in this context?

Although Figure 6E and F are somewhat redundant with Fig. 7A, B I would suggest to keep the data as the assay used is different and shows colonization of different tissues than the lung.

Regarding data quantification, can you please specify the number "n" for how many experiments were performed and the test used to calculate p-values in the respective figure legends? This information is currently incomplete and must be provided in the figure legends. The small white scale bars in Fig. 1C might not be visible in the printed version.

 REFEREE REPORTS

Referee #1:

The authors have made significant revisions to the manuscript. The investigators have included additional patient data sets for clinical validation. The metastasis analysis in mice has been strengthened. Cell cycle has been analyzed and cell polarity characterized more rigorously. Using mutation analysis of seed target sites in MARK4 3'UTR, MARK4 was confirmed as a direct target of miR515-5p. While the manuscript has been improved, there are a few relatively minor points that need attention.

1. The newly included clinical data sets associating miR515-5p with poor outcome in breast and lung cancers are in supplemental files (Supplementary Figure 10). These results are more convincing and are generated from larger patient cohorts compared to the results presented in Figure 6A and B. Therefore, it is more appropriate to have them in the main figure while the current Fig 6A and B can be moved to supplemental. Moreover, it is surprising not to see analysis of MARK4 expression in the new data sets.
2. The MARK4 3'UTR mutation analysis provides evidence for MARK4 as a direct miR515-5p target. This important confirmation would be appropriate in the main figure.
3. The functional analysis of miR515-5p in metastasis in mice has been improved and extended to MARK4 (Figure 7A and B). This analysis is specific to lung metastatic colonization and makes Figures 6E and F redundant. The figures 6E and F do not really depict successful lung colonization.

Referee #2:

The revisions have addressed the biological issues adequately.

We have now made the final changes to the manuscript as suggested. These include:

1. The inclusion of the larger cohort Kaplan Meier curves to Figure 6 and the transfer of the similar data in figure 6 to the Appendix Fig S10. In answer to your question, the dataset used for the miRNA did not contain data for MARK4 and other available dataset did not provide superior data to those already in our manuscript.

2. The Supplementary material is now supplied as a single pdf labelled Appendix and reference to the supplementary figures and table has been modified in the text of the manuscript to conform to the Embo Reports format;
3. Number of technical repeats and repeat experiments has now been added to all figure legends;
4. The RNA-Sea data has now been deposited on a public database and the following sentence added to the Material and Methods section: "The RNA-Seq raw data have been deposited in the ENA database with accession number PRJEB11884."
5. The colour of the scale bar in the microscopy images presented in figure 1C has now been changed to black to help visibility.
6. A synopsis with accompanying illustration has now been generated;
7. We did not move the UTR mutagenesis results from the supplementary figures to the main figure as this further busied an already very full Figure 2 and these data only represented a control that we judged better suited for supplementary material.

3rd Editorial Decision

13 January 2016

I am very pleased to accept your manuscript for publication in the next available issue of EMBO reports. Thank you for your contribution to our journal.